# Time Series Generation Under Data Scarcity: A Unified Generative Modeling Approach

**Tal Gonen**[*] **Itai Pemper**[*] **Ilan Naiman** **Nimrod Berman** **Omri Azencot**
Faculty of Computer and Information Science
Ben-Gurion University of The Negev
{talgon, itaipem, naimani, bermann}@post.bgu.ac.il
azencot@bgu.ac.il

## Abstract

Generative modeling of time series is a central challenge in time series analysis, particularly under data-scarce conditions. Despite recent advances in generative modeling, a comprehensive understanding of how state-of-the-art generative models perform under limited supervision remains lacking. In this work, we conduct the first large-scale study evaluating leading generative models in data-scarce settings, revealing a substantial performance gap between full-data and data-scarce regimes. To close this gap, we propose a unified diffusion-based generative framework that can synthesize high-fidelity time series across diverse domains using just a few examples. Our model is pre-trained on a large, heterogeneous collection of time series datasets, enabling it to learn generalizable temporal representations. It further incorporates architectural innovations such as dynamic convolutional layers for flexible channel adaptation and dataset token conditioning for domain-aware generation. Without requiring abundant supervision, our unified model achieves state-of-the-art performance in few-shot settings—outperforming domain-specific baselines across a wide range of subset sizes. Remarkably, it also surpasses all baselines even when tested on full datasets benchmarks, highlighting the strength of pre-training and cross-domain generalization. We hope this work encourages the community to revisit few-shot generative modeling as a key problem in time series research and pursue unified solutions that scale efficiently across domains. Code is available at https://github.com/azencot-group/ImagenFew.

## 1 Introduction

Many engineering and scientific domains face challenges in collecting high-quality time series data due to cost, privacy, and other barriers. In seismology, earthquake recordings are sparse and geographically limited [57]; climate research requires expensive, long-term sensor deployments [30]; and biomedical data like ECGs often suffer from under-representation and privacy constraints [15, 49]. These limitations hinder the development of robust machine learning models, which typically rely on large, diverse datasets. A promising alternative to large-scale time series collection is the training of generation models [73, 51, 50, 74]. These models aim to capture both the distribution of features at each time step and the complex temporal dynamics across time. Once trained to approximate the underlying data distribution, such models enable the generation of novel, reliable data samples related to the original dataset. While generative modeling offers a promising way to alleviate data scarcity, existing models are often designed and evaluated under the assumption of *abundant training data*. In particular, despite the growing interest in generative approaches, their performance in *low-data* settings, which reflect many real-world scientific applications, remains largely unexplored.

---

[*]Equal Contribution

39th Conference on Neural Information Processing Systems (NeurIPS 2025).

In this context, our first main contribution is a systematic evaluation of state-of-the-art time series generation models under *low-data conditions*. We introduce a novel benchmark that brings together datasets from diverse real-world domains, including finance, climate, and biomedicine, and simulate data-scarce scenarios by limiting training to a small fraction of each dataset (e.g., 5%) [11]. This benchmark allows us to assess the resilience of generative models when data is limited, mirroring the constraints commonly faced in practical applications. Our results reveal a consistent drop in generation quality under such conditions, highlighting that current approaches struggle to maintain good performance with limited data. This finding underscores the pressing need for more robust methods that can operate effectively in low-data regimes.

Existing time series generation models are typically trained separately for each dataset or problem domain [73, 51, 50, 74, 34], a paradigm that proves restrictive in scenarios characterized by limited data availability. Inspired by the success of unified training approaches for vision [39], language [8], and non-generative tasks in time series [24, 11, 70, 17], we extend this paradigm to generative modeling under data-scarce conditions. Rather than training separate models for each dataset, we aim to build a single generative model exposed to a wide range of domains, including energy, finance, climate, and traffic, to encourage cross-domain generalization and few-shot generation. We hypothesize that such diversity enables the model to learn transferable temporal structures, improving its ability to generate high-quality samples even when only a few training examples are available.

To realize this goal, our second main contribution is the development of a *unified generative model* for time series data, trained across a heterogeneous set of domains. We first pre-train our model on a large, diverse time series corpus, and then fine-tune it on downstream generative tasks with limited data. Our architecture builds on the ImagenTime framework [50], which maps time series to invertible image representations, enabling the use of diffusion-based generation in the image domain. To seamlessly handle datasets with varying channel dimensions, we propose a *Dynamic Convolution* (DyConv) layer that interpolates the weights of the channel dimension, ensuring a unified architecture even when test-time inputs differ in channels structure. Finally, to enable domain-specific sampling in our multi-domain trained model, we introduce a *dataset token* mechanism that conditions the diffusion process on domain-specific information.

Our third main contribution is the empirical demonstration of *significant improvements* over existing methods. Our approach maintains strong performance when trained on only 15%, 10%, or even 5% of the original dataset. We further evaluate our model under extreme low-resource conditions, where only 10, 25 and 50 training examples are available, and observe that it continues to produce high-quality samples. Specifically, our unified model achieves, on average, over 47.65% and 23.94% performance gains on Discriminative Score and contextFID, respectively, compared to state-of-the-art baselines across multiple datasets and data-limited scenarios. Beyond performance metrics, we conduct in-depth analyses, including scaling law experiments, to better understand the relationship between data volume, model capacity, and generalization. We believe that our model, extensive empirical evaluation, and the proposed benchmark, can serve as a foundation for further research and drive progress in time series generation under data-constrained environments.

## 2   Related Work

**Generative modeling of time series.**   There are three major existing paradigms for generative modeling of time series data. The first is based on Generative Adversarial Networks (GANs) [26], which have been applied to capture temporal dynamics through adversarial learning [73, 34]. A second line of work employs Variational Autoencoders (VAEs) [38], leveraging probabilistic latent representations to model sequential structure [51, 42, 19]. More recently, the success of diffusion models in domains such as image [28, 62, 61, 63, 4, 36, 6], audio generation [40, 44] and other domains [29, 5, 76] has sparked growing interest in their adaptation to time series [16, 74, 50, 22]. Despite these advances, existing models have largely been developed and evaluated under the assumption of abundant training data. None of these approaches has systematically investigated performance under severe data scarcity, a condition common in many real-world domains such as healthcare, seismology, and climate science. In contrast, our work demonstrates the feasibility of time series generation under extreme low-data conditions, highlighting its practical relevance and robustness in real-world deployment scenarios.

**Unified training of time series.**   Recent advances in foundation models, such as large language models [8, 67, 65] and vision transformers [39, 45], demonstrate the power of training on large-scale,

diverse datasets to enable broad generalization across tasks with minimal fine-tuning [7]. Motivated by this success, researchers have begun exploring analogous approaches in time series analysis, aiming to develop models with similar general-purpose capabilities. One stream of research investigates how to adapt pre-trained language models for time series tasks [77, 27, 10, 35, 13]. Another stream of work focuses on designing specialized architectures that can handle the heterogeneity of time series data [17, 71, 24, 11]. For example, TimesNet [71] extracts frequency-aware representations using Fourier-based transformations to better capture multi-scale temporal patterns. UniTS [24] employs a modified transformer block to capture universal time series representations, enabling transferability from a heterogeneous, multi-domain pre-training dataset, and TimesFM [17] is based on pre-training a decoder-style attention model using a large time series corpus. In spite of the significant progress in time series tasks such as forecasting, classification, and anomaly detection, generative modeling for time series are typically trained on and adapted to individual datasets. In this work, we take a step toward unifying generative modeling through pre-training, introducing a unified model for time series generation. Our goal is to develop a single, versatile model that can generalize across domains and perform effectively on extremely small datasets, requiring only minimal fine-tuning.

## 3 Background

**Problem formulation.** Let a time series be defined as a sequence of real-valued vectors observed at discrete time steps, $\mathbf{x}_{1:T} = (x_1, x_2, \ldots, x_T)$, where each $x_t \in \mathbb{R}^d$. We are given a small dataset $\mathcal{D}_{\text{data}} = \{\mathbf{x}_{1:T}^{(i)}\}_{i=1}^N$ of samples drawn from an unknown distribution $p(\mathbf{x}_{1:T})$. When $N$ is small enough that generative models struggle to learn meaningful temporal structure, we define it as the few-shot setting. This typically corresponds to datasets containing tens of examples, or less than one order of magnitude below the size commonly used to train deep time series models. The goal is to learn a generative model $p_\theta(\mathbf{x}_{1:T})$ such that $p_\theta(\mathbf{x}_{1:T}) \approx p(\mathbf{x}_{1:T})$, despite the limited size of $\mathcal{D}_{\text{data}}$.

**ImagenTime [50]** is a diffusion-based framework for time series generation that leverages advances in high-resolution image synthesis. It first maps time series into 2D images using, e.g., a delay embedding transformation. In delay embedding, a univariate or multivariate time series is mapped into an image by extracting local patches over the temporal axis and organizing them spatially. Mapping a time series to the image domain enables the use of powerful vision diffusion models for generative modeling. After generation in the image space, the samples are mapped back to the time series domain via an inverse delay embedding. This approach bypasses architectural constraints of temporal models and achieves strong performance in both unconditional and conditional generation tasks. In our work, we adopt ImagenTime as the generative backbone due to its high sample fidelity and compatibility with few-shot adaptation.

## 4 Benchmarking Few-Shot Capabilities of Generative Models

We study state-of-the-art time series generators in the *few-shot* regime, where only a limited collection $\mathcal{D}_{\text{data}} = \{\mathbf{x}_{1:T}^{(i)}\}_{i=1}^N$ is available, with small $N$ (from tens to hundreds of examples). Our comparison spans the three major generative paradigms: VAEs, GANs, and diffusion models. We evaluate four strong recent representatives of these approaches: TimeGAN [73], KoVAE [51], DiffusionTS [74], and ImagenTime [50]. All four have achieved impressive results on well-established benchmarks when trained on *abundant data*; here, we test how well those results transfer to *data-scarce* settings.

**Benchmark datasets and evaluation metrics.** To evaluate their performance, we collect a suite of 12 real-world and synthetic datasets covering a wide spectrum of domains, temporal dynamics, and channel dimensionalities: *MuJoCo* [66], *ETTm1*, *ETTm2*, *ETTh2* [75], *Sine*, *Weather* [69], *ILI* [12], *Saugeen River Flow* [47], *ECG200* [53], *SelfRegulationSCP1* [2], *AirQuality* [72], and *StarLightCurves* [56]. We provide detailed statistics and preprocessing steps in App. A.2. To simulate data scarcity, we subsample each dataset using two complementary strategies: *percentage-based* sampling, retaining $5\%$, $10\%$, or $15\%$ of the sequences; and *fixed-count* sampling, limiting the training set to $\{\#10, \#25, \#50\}$ sequences. Unlike genuinely tiny datasets, this design lets us compare generated samples with held-out real data from the same distribution. We assess generation quality with three widely adopted measures for time series: *Discriminative Score* [73], *Predictive Score* [73], and *contextFID* [33]. These metrics respectively quantify (i) the distinguishability of generated from

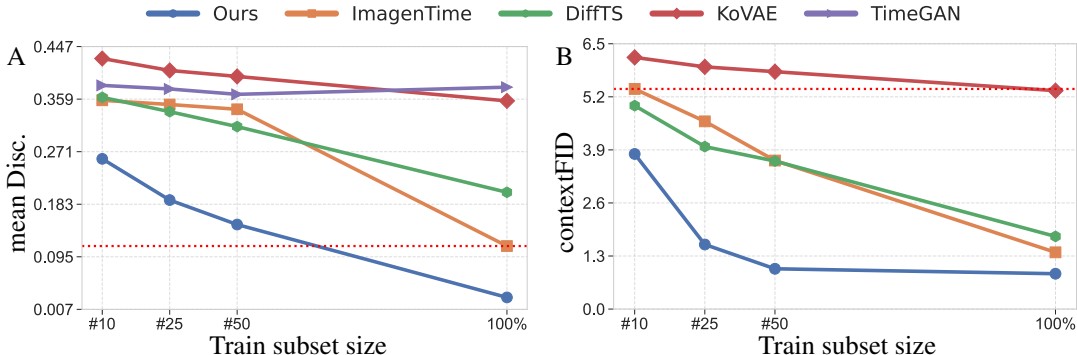

Figure 1: We compare five models in a few-shot scenario using Discriminative Score (A) and contextFID (B), where in both, lower-is-better. Our model consistently outperforms others across all subset sizes, with clear gains on smaller subsets. TimeGAN consistently reports contextFID scores above 6.5 across all subset sizes, which is why it is not visible in the visualization shown in Fig. (B).

real samples, (ii) the preservation of temporal dependencies important for prediction, and (iii) the proximity of contextual embeddings between synthetic and real sequences. Formal definitions and implementation details are given in App. B.3.

**Results.** We train all models across all setups (one model per subset) and present the partial results in Fig. 1. The plots show that state-of-the-art (SOTA) models experience significant performance degradation in data-scarce settings. More comprehensive results are provided in Tab. 1. These findings confirm that few-shot generation remains a major challenge for current SOTA models. A commonly proposed approach to mitigate data scarcity is data augmentation [37, 60]. However, in the context of time series, augmentation techniques may harm performance when the underlying distributional biases are not aligned [68, 52]. Instead, we draw inspiration from the promising performance of unified training frameworks in zero- and few-shot time-series forecasting [24, 11, 70, 17]. Building on these advances, we extend the unified-training paradigm to the generative setting and develop a model $p_\theta(\mathbf{x}_{1:T} \mid \mathcal{D}_{\text{data}})$ that can effectively adapt to the few-shot target distribution and approximate $p(\mathbf{x}_{1:T})$ for novel time series generation tasks.

## 5 Our Unified Generative Time Series Model

Existing methods for time series generation typically train a dedicated model for each specific dataset or domain [73, 51, 50, 74, 34]. While effective when abundant training data is available, such strategies become impractical in low-data settings, where the low number of examples limits the model's ability to produce high-quality samples (Sec. 4). To address this, we propose a unified generative modeling framework that trains a single model across a heterogeneous collection of time series datasets spanning multiple domains, including finance, energy, climate, traffic, and biomedical signals, see App. A.1. The key idea is that exposure to a broad range of temporal structures and statistical properties during pre-training enables the model to learn domain-agnostic representations of time series dynamics. This, in turn, improves the model's ability to generate high-quality samples even when only a few examples from a new target domain are available. Analogous to the way foundation models in vision and language benefit from large-scale, diverse pre-training [55, 8, 54], we hypothesize that a unified time series generative model can generalize more effectively under few-shot conditions by leveraging prior knowledge acquired from diverse sources.

**Modeling.** Similarly to ImagenTime [50], our approach builds upon an image-based framework [36]. We adopt this image-based generative backbone for several key reasons: (1) it achieves state-of-the-art results in unconditional and conditional time series generation; (2) it exhibits strong generalization capabilities across a wide range of sequence lengths, including very long horizons; and (3) it benefits directly from rapid progress in the vision diffusion community, allowing continual improvements in generative quality through better backbone models. We illustrate the core components of our generative modeling framework in Fig. 2. For each sample in the train set, $\mathbf{x}_{1:T} = (x_1, x_2, \ldots, x_T)$, where each $x_t \in \mathbb{R}^d$ and $d$ is the channel number, we convert it through a delay embedding

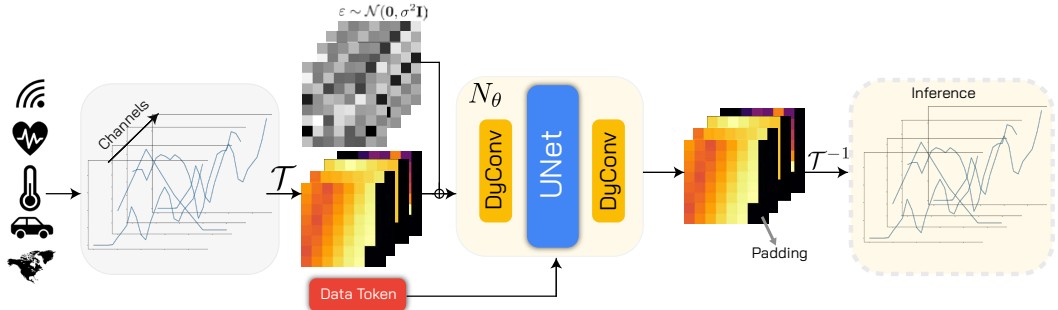

Figure 2: In our architecture, time series data from a wide range of domains are first transformed into image representations. Then, their noisy version and their data tokens are processed by a neural network equipped with dynamic convolutions (DyConvs), which accommodate varying channel sizes. Finally, only during inference, the generated images are transformed back into time series data.

transformation, $\mathcal{T}$, into an image $\mathcal{T}(\mathbf{x}_{1:T}) = x_{\text{img}}^0 \in \mathbb{R}^{C \times H \times W}$. We additionally pad the resulting image to a square shape. We follow EDM [36] and ImagenTime [50] frameworks, adding noise $\varepsilon \sim \mathcal{N}(0, \sigma^2 I)$ to $x_{\text{img}}^0$, $x_{\text{img}}^\sigma = x_{\text{img}}^0 + \varepsilon$, where $\sigma$ is the noise schedule. Then, the objective of the model is to clean the noise via the following loss function:

$$\mathcal{L} = \mathbb{E}_{\sigma, x_{\text{img}}^0, \varepsilon}\left[\lambda(\sigma) \left\| \Gamma_\theta(x_{\text{img}}^\sigma; \sigma) - x_{\text{img}}^0 \right\|_2^2\right] , \tag{1}$$

where $\lambda(\sigma)$ is a weighting function and $\Gamma_\theta$ is defined as follows:

$$\Gamma_\theta(x_{\text{img}}^\sigma; \sigma) = c_{\text{skip}}(\sigma)x_{\text{img}}^\sigma + c_{\text{out}}(\sigma)N_\theta(c_{\text{in}}(\sigma)x_{\text{img}}^\sigma; c_{\text{noise}}(\sigma); y) , \tag{2}$$

where $N_\theta$ is a neural network. $c_{\text{skip}}$ modulates the skip connection, $c_{\text{in}}$ and $c_{\text{out}}$ scale the input and output magnitudes, and $c_{\text{noise}}$ maps noise level $\sigma$ into a conditioning input for $N_\theta$. For robust training, we use the same values as in EDM's preconditioning [36]. We feed the network with an additional optional dataset token input $y$, explained next. Unlike [50], we apply a *dynamic binary mask* to the padded input sequences. The mask is adjusted at runtime based on the actual sequence length, marking valid time steps and excluding padded regions. This allows the model to distinguish meaningful data from padding and ensures that, during training, it learns to focus exclusively on valid regions while effectively ignoring padded values. This mechanism enables the model to be pre-trained on sequences of a fixed length and subsequently fine-tuned on sequences of different lengths, supporting *time series generation of variable-length inputs* without architectural modifications or additional pre-training. In contrast to traditional approaches [50, 73, 74], which typically rely on fixed-length inputs or require retraining for each length, our method naturally supports cross-temporal resolution transfer. Additional implementation details of the masking mechanism are provided in Appendix B.1. This capability is empirically validated by our experiments in Section 6.2, which show that models pre-trained on one sequence length maintain strong performance when fine-tuned on different lengths.

**DyConv.** Unlike prior generative time series models, our model must flexibly handle varying input and output channel sizes. A common workaround involves setting a maximum number of channels and padding all samples to match this size—similar to how we standardized the time axis. While simple, this approach (1) restricts the few-shot training regime since the input channel is constrained to a fixed size, and (2) wastes computation on datasets with fewer channels. We address this by *DyConv*, a dynamic convolutional layer for adaptive channel handling. Inspired by DyLinear [24], we use a single learnable canonical kernel of shape $[K, K, C_0, C_1]$, where $K$ is the kernel size and $C_0, C_1$ are fixed reference dimensions. At runtime, the kernel is resized via bicubic interpolation to match each dataset's actual channel dimensions. Formally, let the input be $x \in \mathbb{R}^{C_{\text{in}} \times H \times W}$ and the target channel size $C_{\text{out}}$. DyConv constructs a kernel $W_{\text{interp}} \in \mathbb{R}^{K \times K \times C_{\text{in}} \times C_{\text{out}}}$ via bicubic interpolation over the channel dimensions of a canonical kernel $W$:

$$\text{DyConv}(x; W) = \text{Conv2D}\left(x, \text{Interp}(W, C_{\text{in}}, C_{\text{out}})\right) , \tag{3}$$

with a similar interpolation applied to the bias. Unlike standard convolutions, DyConv allows a single parameter set to generalize across datasets with varying input/output channels. In our UNet, DyConv is used both to map inputs to a shared embedding space and to project outputs back to the original dimensionality. This enables handling of multivariate time series with differing variable

counts, without architecture changes. In Sec. 6.5, we show that training with DyConv significantly outperforms naïve channel padding, offering improved efficiency and avoiding the constraints imposed by padding. Further details and ablation results are provided in App. B.1 and App. C.1.

**Dataset token.**    To enable multi-domain sampling, we introduce a *dataset token* mechanism that conditions the diffusion process on domain-specific information. Each dataset is assigned a unique token that acts as an identifier of its source domain. During both training and generation, this token is mapped to a learnable embedding and injected into the denoising network via the adaptive group normalization (AdaGN) module [20]. This embedding is incorporated into the intermediate features of the diffusion model, allowing it to modulate its behavior based on the dataset context. Through this mechanism, the model captures dataset-specific characteristics while still leveraging shared temporal patterns across domains. At inference time, the dataset token continues to guide the generative process, ensuring that samples are drawn from the correct target distribution corresponding to the desired domain. Additionally, we experiment with training and fine-tuning without the data token. Interestingly, while pre-training without the dataset token is still effective when fine-tuning occurs, we cannot reliably sample from the pre-trained model without a fine-tuning stage, see Sec. 6.5.

**Pre-training.**    We pre-train our model across a wide collection of time series datasets. This unified pre-training exposes the model to a broad range of temporal structures and input dimensionalities, encouraging the learning of transferable representations that are useful in data-scarce settings. Specifically, we utilize the datasets: Stocks, Energy [9], ETTh1 [75], Exchange [41], MSL [32], SMAP [32], PSM [1], SMD [64], ECG5000 [25], NonInvasiveFetalECGThorax1 [59], SelfRegulationSCP2 [2], Blink [14], ElectricDevices [43], Trace [58], FordB [18], UWaveGestureLibrary [46], EMOPain [21], Chinatown [18], and SharePriceIncrease [48]. These datasets combine commonly used collections from various tasks in UniTS [24] with datasets from generative modeling research [50, 74]. In our approach, all of these datasets are jointly employed during the pre-training phase, encompassing approximately 300,000 time series in total. During pre-training, we utilize the full corpus over 1,000 epochs with a learning rate of $10^{-4}$, conducted in a distributed setup across two NVIDIA RTX 4090 GPUs, requiring roughly 4 hours of training time.

**Fine-tuning for few-shot generation.**    To address data scarcity in novel domains, we fine-tune the model with a dedicated dataset token for each new dataset encountered. This token is mapped to a learnable embedding that represents the identity of the new domain. The embedding is initialized randomly and optimized jointly with the rest of the model during fine-tuning. It serves as a signal to distinguish the target dataset from those seen during pre-training, thereby guiding the model to generate samples that reflect the unique characteristics of the new distribution. Fine-tuning is performed on a small subset of the target dataset, in accordance with our few-shot benchmark protocol. This enables the model to quickly adapt its generative behavior to the new domain, supporting strong generalization in low-resource scenarios. We also experimented with other fine-tuning strategies, such as freezing all weights except the biases or applying Low-Rank Adaptation (LoRA) [31]. However, we observe poor results compared to our fine-tuning, see a detailed analysis in App. C.2.

## 6    Experiments

In this section, we empirically evaluate the performance and key aspects of our model. We start by evaluating its performance on the novel few-shot generation benchmark against the SOTA baselines (Sec. 6.1). Then, we also show our robustness when evaluated on varying-length data (Sec. 6.2). Next, we explore the effects of model scale and its generalization abilities to longer sequences (Sec. 6.3). Finally, we perform ablation studies on the main properties of our method (Sec. 6.5).

### 6.1    Few-Shot Benchmark for Time Series Generation

To assess the effectiveness of our unified generative model in low-data regimes, we conduct a comprehensive few-shot generation study using the same target datasets introduced in Sec. 4. Our pre-trained model is fine-tuned on each dataset and evaluated against the baselines from Sec. 3: ImagenTime [50], DiffusionTS [74], KoVAE [51], and TimeGAN [73]. We report the averaged Discriminative Score (Disc.), Predictive Score (Pred.), and contextFID (c-FID) in Tab. 1. Across all subset sizes and evaluation metrics, our method consistently outperforms the baselines, demonstrating strong performance in both percentage-based and count-based few-shot settings. Remarkably, our

Table 1: Full and few-shot results across subset scales. We report Discriminative Score (Disc.), Predictive Score (Pred.), and contextFID (c-FID)↓. For each subset size, values reflect averages across all evaluation datasets. Our method, ImagenFew (pre-trained then fine-tuned on sequence length 24), consistently outperforms all baselines. Bold marks best, underline second-best.

| Subset | Metric | ImagenFew | ImagenTime | DiffTS | KoVAE | TimeVAE | TimeGAN | Improvement |
|--------|--------|-----------|------------|--------|--------|---------|---------|-------------|
| 100% | Disc. | **0.027** | 0.113 | 0.203 | 0.356 | 0.185 | 0.379 | 76.1% |
|  | Pred. | **0.448** | 0.452 | 0.457 | 0.482 | 0.455 | 0.520 | 0.88% |
|  | c-FID | **0.866** | 1.390 | 1.78 | 5.351 | 3.068 | 7.912 | 37.69% |
| 5% | Disc. | **0.110** | 0.321 | 0.248 | 0.389 | 0.239 | 0.388 | 53.97% |
|  | Pred. | **0.458** | 0.469 | 0.479 | 0.485 | 0.467 | 0.499 | 1.93% |
|  | c-FID | **0.674** | 3.464 | 2.087 | 5.232 | 1.747 | 11.055 | 61.42% |
| 10% | Disc. | **0.083** | 0.248 | 0.233 | 0.375 | 0.232 | 0.402 | 64.22% |
|  | Pred. | **0.452** | 0.458 | 0.469 | 0.491 | 0.466 | 0.494 | 1.31% |
|  | c-FID | **0.578** | 2.757 | 1.944 | 5.326 | 1.692 | 14.057 | 65.84% |
| 15% | Disc. | **0.066** | 0.236 | 0.229 | 0.363 | 0.215 | 0.384 | 69.30% |
|  | Pred. | **0.451** | 0.458 | 0.464 | 0.484 | 0.465 | 0.486 | 1.52% |
|  | c-FID | **1.086** | 2.211 | 2.064 | 5.644 | 1.303 | 8.290 | 16.66% |
| #10 | Disc. | **0.259** | 0.357 | 0.362 | 0.427 | 0.312 | 0.382 | 16.99% |
|  | Pred. | 0.489 | 0.492 | 0.525 | 0.514 | 0.510 | **0.467** | -4.71% |
|  | c-FID | 3.800 | 5.393 | 4.984 | 6.166 | **2.383** | 28.572 | -59.46% |
| #25 | Disc. | **0.190** | 0.350 | 0.338 | 0.407 | 0.260 | 0.376 | 43.78% |
|  | Pred. | **0.467** | 0.469 | 0.498 | 0.515 | 0.473 | 0.538 | 0.42% |
|  | c-FID | **1.582** | 4.599 | 3.980 | 5.934 | 1.874 | 7.277 | 15.58% |
| #50 | Disc. | **0.149** | 0.342 | 0.313 | 0.397 | 0.239 | 0.367 | 37.66% |
|  | Pred. | **0.460** | 0.466 | 0.493 | 0.502 | 0.475 | 0.472 | 1.28% |
|  | c-FID | **0.987** | 3.639 | 3.626 | 5.816 | 1.75 | 9.744 | 43.60% |

unified model, even when fine-tuned on only 5% of the data, outperforms ImagenTime [50] trained on the full dataset and achieves average performance gains of 55.72% and 54.25% in discriminative and contextFID scores, respectively, over all competing models. Beyond few-shot scenarios, our approach also excels in data-rich settings. Fine-tuning our pre-trained model on large datasets yields further performance gains, highlighting the benefits of pre-training across diverse domains. This suggests that our model acquires a robust inductive bias that enables superior generalization compared to models trained from scratch. These findings underscore the versatility and strength of our unified generative framework, particularly in few-shot generation. However, some datasets, such as Weather [69] and ECG200 [2], remain challenging under limited data conditions, as seen in the full results table provided in App. C.6.

## 6.2 Pre-training with Varying Sequence Lengths

We examine how the pre-training sequence length influences downstream fine-tuning performance. We pre-trained four models using fixed sequence lengths of 12, 24, 36, and 64 time steps, following the protocol described in Sec. 5. Each model was fine-tuned and evaluated on six datasets: ECG200, ETTh2, ETTm1, ETTm2, ILI, and a synthetic sine wave, using sequence lengths of 12, 24, 36, and

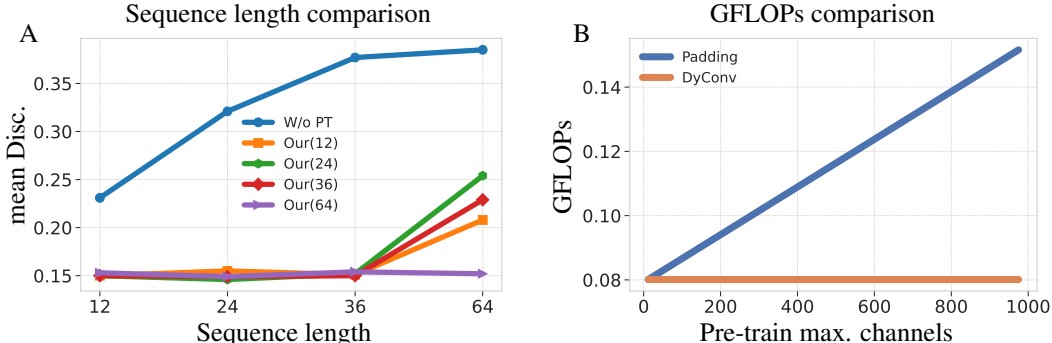

Figure 3: (A) Length generalization capabilities comparing five models, including four pre-trained on fixed sequence lengths and one without pre-training (w/o PT). Our(t) models were pre-trained exclusively on sequences with t steps. (B) Computational complexity measured in GFLOPs.

64. For comparison, we include models trained from scratch (w/o PT baseline). Performance is reported as the average discriminative score under a few-shot generation setup using 5%, 15%, #25, and #50 of the available samples. Results are shown in Fig. 3A. On sequence lengths of 12, 24, and 36, all pre-trained models achieve consistently strong performance, significantly outperforming the baseline. As sequence length increases and thus, making the task more difficult, the baseline performance degrades markedly, whereas pre-trained models remain robust. At length 64, the model pre-trained on that specific length substantially outperforms those pre-trained on shorter sequences. These results demonstrate that our method not only generalizes well in low-data regimes, but also transfers effectively across different temporal resolutions, an essential property for real-world time series applications.

## 6.3 Pre-training Impact Across Model Scales

We now explore the influence of model size on fine-tuning performance, focusing on the more challenging setup introduced in the previous section, where we observed strong results on generating time series of length 64. To this end, we pre-train four models with increasing parameter counts: Base (6M), Medium (15M), Large (26M), and XL (40M), using an identical pre-training protocol. Each model is subsequently fine-tuned and evaluated across all downstream tasks, with the generation target fixed to 64 time steps. For comparison, we also include models trained without pre-training ("w/o PT"). We report Discriminative Score (Disc.), Predictive Score (Pred.), and contextFID (c-FID), with all results averaged over datasets and subset-size conditions to reveal general performance trends. This setup enables us to assess how model capacity and pre-training interact in the few-shot regime.

Tab. 2 summarizes the effect of model size on fine-tuning performance for 64-step time series generation in the few-shot setting. Several key trends emerge: (1) Fine-tuned models consistently outperform their w/o PT counterparts across all scales. (2) Discriminative Score improves with model size under both training modes, with the fine-tuned Large model achieving the best score (0.130). Interestingly, the XL model performs slightly worse than the Large model, suggesting diminishing re-

Table 2: Model size ablation comparing fine-tuned (FT) models with those trained without pre-training (w/o PT).

| | | Base | Medium | Large | XL | XL→S Det. |
|---|---|---|---|---|---|---|
| Disc. | FT | 0.15 | 0.16 | **0.13** | 0.13 | -14.02% |
| | w/o PT | 0.35 | 0.35 | 0.26 | 0.27 | -23.29% |
| c-FID | FT | 4.85 | 5.03 | 4.56 | **4.51** | -7.37% |
| | w/o PT | 15.87 | 15.36 | 8.12 | 8.15 | -48.62% |
| Pred. | FT | 0.4993 | 0.4974 | **0.4959** | 0.4965 | -0.57% |
| | w/o PT | 0.5043 | 0.5036 | 0.5007 | 0.5023 | -0.40% |

turns at higher capacity. (3) c-FID follows a similar trend, with pre-training yielding substantial gains, especially for smaller models. (4) Predictive Scores remain relatively stable across scales, with a slight improvement for larger pre-trained models. (5) Pre-training also reduces the sensitivity of performance to scale, as shown by the XL→S Det. column. For example, c-FID in w/o PT deteriorates by 48.62% when moving from XL to Base, while fine-tuned models show much smaller gaps. Overall, pre-training allows smaller models to approach the performance of larger ones.

Table 3: Out-of-domain fine-tuning. The models are tested on a sequence length of 24.

| | | Disc. ↓ | | | | | | c-FID ↓ | | | | | |
|---|---|---|---|---|---|---|---|---|---|---|---|---|---|
| Dataset | Method | 5% | 10% | 15% | 10# | 25# | 50# | 5% | 10% | 15% | 10# | 25# | 50# |
| Stocks | ImagenFew | **0.018** | **0.017** | **0.018** | **0.138** | **0.119** | **0.054** | **0.132** | **0.090** | **0.054** | **0.379** | **0.430** | **0.183** |
| | ImagenTime | 0.204 | 0.306 | 0.316 | 0.275 | 0.206 | 0.247 | 0.846 | 0.742 | 0.691 | 3.267 | 2.616 | 1.845 |
| Exchange | ImagenFew | **0.046** | **0.036** | **0.044** | **0.352** | **0.279** | **0.178** | **0.137** | **0.119** | **0.124** | **2.574** | **2.222** | **0.833** |
| | ImagenTime | 0.470 | 0.470 | 0.473 | 0.496 | 0.476 | 0.491 | 3.552 | 3.366 | 3.285 | 8.132 | 5.901 | 4.639 |

## 6.4 Out-of-domain Generalization

We evaluate our model's ability to generalize to unseen domains. To achieve this, we intentionally excluded financial datasets (Stocks and Exchange) from the pre-training phase. Subsequently, we fine-tuned our model, ImagenFew, separately on each of these out-of-domain datasets. The results, compared against the ImagenTime baseline, are presented in Table 3. In this setting, we find that ImagenFew significantly outperforms ImagenTime across all metrics and experimental setups on both the Stocks and Exchange datasets. This strong performance is particularly noteworthy given that our model had no prior exposure to financial data during pre-training. This suggests our unified approach is a promising step toward cross-domain generalization, where knowledge from source domains can be effectively transferred to a new, unseen one.

Table 4: Ablation study on the effect of DyConv. We compare against a padding-based baseline with equivalent capacity (6M parameters). Results are averaged across all evaluation datasets.

| Metric | 5% | | 10% | | 15% | | #10 | | #25 | | #50 | |
|---|---|---|---|---|---|---|---|---|---|---|---|---|
| | ImagenFew | Baseline | ImagenFew | Baseline | ImagenFew | Baseline | ImagenFew | Baseline | ImagenFew | Baseline | ImagenFew | Baseline |
| Disc. ↓ | **0.110** | 0.291 | **0.083** | 0.250 | **0.066** | 0.240 | **0.259** | 0.381 | **0.190** | 0.372 | **0.149** | 0.355 |
| Pred. ↓ | **0.458** | 0.462 | **0.452** | 0.460 | **0.451** | 0.458 | **0.489** | 0.489 | **0.467** | 0.474 | **0.460** | 0.466 |
| c-FID ↓ | **0.674** | 4.190 | **0.578** | 4.864 | **1.086** | 3.532 | **3.800** | 7.889 | **1.582** | 5.940 | **0.987** | 5.905 |

Table 5: Few-shot evaluation with and without dataset token conditioning.

| Metric | 5% | | 10% | | 15% | | #10 | | #25 | | #50 | |
|---|---|---|---|---|---|---|---|---|---|---|---|---|
| | Cond. | Uncond. | Cond. | Uncond. | Cond. | Uncond. | Cond. | Uncond. | Cond. | Uncond. | Cond. | Uncond. |
| Disc. ↓ | 0.11 | 0.10 | 0.08 | 0.08 | 0.06 | 0.06 | 0.25 | 0.25 | 0.19 | 0.18 | 0.14 | 0.15 |
| Pred. ↓ | 0.45 | 0.45 | 0.45 | 0.45 | 0.45 | 0.45 | 0.48 | 0.49 | 0.46 | 0.46 | 0.46 | 0.46 |
| c-FID ↓ | 0.67 | 0.79 | 0.57 | 0.55 | 1.08 | 1.16 | 3.80 | 2.42 | 1.58 | 1.23 | 0.98 | 0.98 |

## 6.5 Ablation of Main Properties

We investigate the effect of our dynamic channel adaptation module (DyConv) and dataset token. We begin by evaluating whether DyConv is necessary for handling varying input sizes and improving few-shot performance. In the App. C.1, we analyze how its internal configuration, specifically, the number of input and output channels, affects generative quality.

**DyConv vs. padded baseline.** To evaluate the necessity of DyConv, we conduct an ablation study in which the module is removed and replaced with static channel padding. Specifically, all input sequences are zero-padded to match the maximum channel dimensionality observed during pre-training. The remainder of the architecture, including the dataset token mechanism, is left unchanged, and the total number of parameters is matched to our main model. The padded baseline is pre-trained using the same procedure described in Sec. 5. To ensure a fair comparison, we introduce a masking mechanism in the loss function such that the denoising network is only penalized for predicting valid (unpadded) dimensions. This isolates the effect of DyConv from other architectural factors. The results, averaged across all evaluation datasets, are presented in Tab. 4. Our model consistently outperforms the padded baseline across all metrics, with especially large gains in Discriminative Score and contextFID. These results demonstrate that DyConv is not only effective but essential. Beyond enabling the handling of varying channel sizes, DyConv contributes a significant performance boost, making it a critical component of our architecture. In contrast, padding-based approaches are inherently constrained by the fixed dimensionality used during pre-training, limiting their ability to generalize in real-world, multi-domain scenarios where input structures vary significantly.

**Effect of dataset token.** We analyze the impact of dataset token guidance during both pre-training and fine-tuning. Table 5 compares models fine-tuned with and without the token (Cond. vs. Uncond.) across all benchmark datasets. Interestingly, both variants achieve similar downstream performance, indicating that once the model is adapted to a specific dataset, the token becomes largely redundant for fine-tuning. This suggests that the model acquires domain-agnostic temporal representations during pre-training, which transfer effectively under limited supervision. In contrast, the dataset token plays a central role during pre-training. It enables the model to learn separate distributional modes across multiple datasets, allowing us to reliably sample from each distribution and evaluate the quality of pre-training. It also allows users to leverage the pre-trained model as an invested resource, enabling targeted sampling without additional fine-tuning. To quantify this effect, we compare the generation performance on the pre-training datasets when models are trained with and without the token. Without the token, the model samples from a mixture of all training distributions, leading to lower-quality and less coherent generations. Averaged across all pre-training datasets, the Discriminative Score and context-FID drop from 0.193 and 4.84 (with token) to 0.252 and 11.07 (without token), as shown in Table 21. This demonstrates the key role of the token in resolving distributional ambiguity during pre-training, enabling domain-specific sampling and improving generation fidelity. While the token has little impact during fine-tuning on a single dataset, it remains essential for both evaluating pre-training and enabling controlled generation in mixed or unseen data settings.

## 6.6 Runtime and Memory Consumption

In our model, DyConv accommodates varying channel sizes while maintaining complete independence between the current sequence's channel size and the pre-train maximum channel size. This

design stands in contrast to the naïve padding-based approach, which inherently introduces a dependency between the two. In this section, we evaluate the computational cost of processing a single sequence of length 24 with the average channel number encountered during pre-training under both setups, as a function of the maximum number of channels used during pre-training. Fig. 3B illustrates a linear correlation between the number of pre-training maximum channels and the computational cost (in GFLOPs) for the naïve padding solution. In contrast, DyConv exhibits a constant computational cost regardless of the channel count. In a single pre-training iteration, as described in Sec. 5, with a batch size of 2048 on an NVIDIA A6000 GPU, DyConv consumes approximately 0.18 GB of memory on average. Under the same setup, the naïve padding approach consumes 2.01 GB, representing a substantially higher memory footprint.

# 7 Conclusion

Recent literature has introduced multiple approaches for time series generation based on GANs, VAEs, and Diffusion models. However, our research demonstrates that the performance of these models deteriorates significantly as the number of available training samples decreases, a common constraint in real-world time series applications where generative modeling is invaluable. This paper addresses this limitation by proposing a two-stage modeling framework. The first stage involves unified pre-training across multiple datasets, followed by a second stage where the model is fine-tuned for specific generative tasks under data-scarce conditions. In contrast to conventional approaches where models are trained on individual datasets, our method exposes the model to a diverse range of temporal structures and statistical properties during pre-training. This exposure facilitates the learning of domain-agnostic representations of time series dynamics. We validate the efficacy of our methodology through a novel benchmark specifically designed for this task and systematically analyze its key properties through comprehensive ablation studies. Our unified generative model achieves state-of-the-art performance in few-shot time series generation, outperforming all baselines by over 55% in discriminative and contextFID scores, even when fine-tuned on just 5% of the data. Additionally, we demonstrate strong robustness across sequence lengths and model scales, with DyConv and dataset token guidance proving essential for efficiency and generalization in real-world, multi-domain scenarios. In conclusion, we believe that this approach has the potential to facilitate the development of large pre-trained models for time series generation, similar to the transformative advancements witnessed in image and language domains.

# 8 Broader Impact

Our framework advances time-series generation by pre-training a unified model across multiple datasets, promising significant gains in data synthesis, especially in data-scarce scenarios. For example, using our approach, geophysicists can generate more high-fidelity examples of earthquake waveforms in locations with rare seismic events, improving hazard assessment models. In healthcare, researchers could generate synthetic, yet realistic, electronic health records to study disease progression without compromising patient privacy, or augment datasets for training diagnostic models for rare diseases where real-world data is limited.

At the same time, because the model spans high-stakes domains like finance and biomedical data, its power to generate high-fidelity sequences also heightens significant societal risks. The potential for misuse of synthetic data is substantial; for instance, the same technology that helps medical researchers could be used to create fraudulent clinical trial data, potentially leading to the approval of ineffective treatments. Beyond direct misuse, a critical danger lies in the overreliance on generated samples, as models trained on augmented datasets may become less robust to the noisy, unpredictable nature of real-world data. Furthermore, there is a risk of downstream model degradation due to hallucinations—the generative model may produce data with subtle, unrealistic artifacts that, while plausible-looking, cause subsequent models to fail in unforeseen ways.

## Acknowledgments

This research was partially supported by the Lynn and William Frankel Center of the Computer Science Department, Ben-Gurion University of the Negev, an ISF grant 668/21, an ISF equipment grant, and by the Israeli Council for Higher Education (CHE) via the Data Science Research Center, Ben-Gurion University of the Negev, Israel.

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

# A    Additional Dataset Details

We provide detailed descriptions of the datasets used in our framework, divided into two groups: those used for large-scale **pre-training** and those used for **evaluating** few-shot generation. The datasets span a broad range of domains, temporal structures, and channel dimensionalities, and include both real-world and synthetic sources. These datasets originate from a variety of source tasks, including classification, forecasting, and anomaly detection. In our setting, however, we do not make use of the task-specific labels or objectives. Instead, we treat all datasets uniformly as raw multivariate time series, enabling a consistent generative modeling framework across diverse data types.

| Name | # Samples | Task | # Variables | Domain |
|---|---|---|---|---|
| Stock | 3661 | Generation | 6 | Finance |
| Energy | 19711 | Generation | 28 | Energy |
| ETTh1 | 8617 | Forecasting | 7 | Energy |
| Exchange | 5288 | Forecasting | 8 | Finance |
| MSL | 58294 | Anomaly Detection | 55 | Space |
| SMAP | 132458 | Anomaly Detection | 25 | Space |
| PSM | 135160 | Anomaly Detection | 25 | Cloud |
| SMD | 7084 | Anomaly Detection | 38 | Cloud |
| ECG5000 | 200 | Classification | 1 | ECG |
| NonInvasiveFetalECGThorax1 | 120 | Classification | 1 | ECG |
| SelfRegulationSCP2 | 500 | Classification | 7 | EEG |
| Blink | 1800 | Classification | 4 | EEG |
| ElectricDevices | 8926 | Classification | 1 | Sensors |
| Trace | 100 | Classification | 1 | Sensors |
| FordB | 3636 | Classification | 1 | Sensors |
| UWaveGestureLibrary | 2238 | Classification | 3 | Human Activity |
| EMOPain | 968 | Classification | 30 | Human Activity |
| Chinatown | 20 | Classification | 1 | Traffic |
| SharePriceIncrease | 965 | Classification | 1 | Finance |

Table 6: Summary of pre-training datasets including number of samples, task type, number of variables, and domain. All datasets are framed as generation tasks.

## A.1    Pre-training Datasets

The following datasets are used for unified pre-training of the diffusion model described in Sec. 5. These datasets cover a wide variety of domains, sequence characteristics, and input sizes, allowing the model to learn generalizable representations. A summary of the pre-training datasets, including the number of samples, original task type, channel size, and domain, is provided in Table 6.

**Stocks.**    The Stocks dataset [73] consists of daily historical Google stock data from 2004 to 2019, comprising six channels: high, low, opening, closing, adjusted closing prices, and trading volume. The data is largely non-periodic and exhibits random-walk behavior typical of financial time series.

**Energy.**    The Energy dataset is a multivariate appliance energy prediction dataset [9], featuring 28 channels with correlated features. It exhibits noisy periodicity and contains continuous-valued measurements typical of real-world energy consumption data.

**ETTh1.**    ETTh1 is part of the Electricity Transformer Temperature (ETT) dataset collection [75], described in App. A.2. It provides hourly readings of transformer load and oil temperature, and is characterized by strong periodic and seasonal patterns.

**Exchange.**    The Exchange dataset [41] contains daily exchange rates of eight foreign currencies - Australia, United Kingdom, Canada, Switzerland, China, Japan, New Zealand, and Singapore, spanning the years 1990 to 2016. This dataset captures long-term financial trends and exhibits moderate temporal variability.

**MSL.** The MSL dataset [32] includes labeled anomalies from the Mars Science Laboratory Curiosity rover. It features both point and contextual anomalies identified from spacecraft telemetry.

**SMAP.** The SMAP dataset [32] contains labeled telemetry anomalies from NASA's Soil Moisture Active Passive satellite. Anomalies are categorized as point or contextual, based on their temporal characteristics.

**PSM.** The PSM dataset [1] contains multivariate server telemetry data collected from multiple application nodes at eBay. It consists of anonymized system-level metrics and is used to evaluate asynchronous multivariate time series anomaly detection methods.

**SMD.** The SMD dataset [64] consists of multivariate monitoring data collected over five weeks from production server machines in a large Internet company. Each machine is treated as a separate entity with its own training and test split and labeled anomalies. The dataset is commonly used to evaluate anomaly detection under operational server conditions.

**ECG5000.** The ECG5000 dataset [25] is sourced from PhysioNet and contains pre-processed heartbeat segments extracted from a long ECG recording of a patient with severe congestive heart failure. Each heartbeat is resampled to a fixed length and annotated into one of five classes using automated labeling.

**NonInvasiveFetalECGThorax1.** The NonInvasiveFetalECGThorax1 dataset [59] from PhysioNet contains abdominal ECG recordings used for noninvasive fetal monitoring. Signals were collected from the thorax of pregnant subjects and are used to study fetal QRS detection in the presence of dominant maternal ECG signals.

**SelfRegulationSCP2.** The SelfRegulationSCP2 dataset [2] contains EEG recordings from an artificially respirated ALS patient performing a brain–computer interface (BCI) task. The subject was instructed to regulate slow cortical potentials to move a cursor up or down, guided by visual and auditory cues. Each trial consists of a 4.5-second segment sampled at 256 Hz.

**Blink.** The Blink dataset [14] contains EEG recordings collected during a binary classification task involving short and long blinks. Each subject was instructed to blink for 2 seconds, with data collected across multiple trials. The EEG was sampled at 255 Hz over a 2-second window per blink.

**ElectricDevices.** The ElectricDevices dataset [43] was collected as part of the UK's Powering the Nation study, aimed at analyzing household electricity usage to inform carbon reduction strategies. It contains electricity consumption readings from 251 households, sampled at two-minute intervals.

**Trace.** The Trace dataset [58] is a synthetic 4-class subset derived from the Transient Classification Benchmark, simulating instrumentation failures in a nuclear power plant. It was originally designed to support research in diagnostic systems for industrial processes. All instances are z-normalized and interpolated to a uniform length.

**FordB.** The FordB dataset [18] originates from a classification challenge at the IEEE World Congress on Computational Intelligence (2008). It contains engine noise recordings used to detect the presence or absence of a specific automotive subsystem symptom. Training data were collected under normal operating conditions, while test data include added noise.

**UWaveGestureLibrary.** The UWaveGestureLibrary dataset [46] contains accelerometer recordings of eight predefined gestures captured along the X, Y, and Z axes. The dataset was collected for personalized gesture recognition using mobile devices.

**EMOPain.** The EMOPain dataset [21] includes motion capture and physiological recordings from participants—both healthy individuals and those with chronic lower back pain—performing physical exercises that mimic daily activities. The dataset supports research in multimodal pain recognition and behavioral analysis.

**Chinatown.** The Chinatown dataset [18] contains time series of pedestrian counts recorded in Melbourne's Chinatown-Swanston Street (North) throughout 2017. Each series represents a day's pedestrian volume, and the task is to classify whether the day is a weekday or a weekend. The dataset was collected as part of a city-wide initiative to monitor and plan urban foot traffic.

**SharePriceIncrease.** The SharePriceIncrease dataset [48] consists of 60-day time series representing daily percentage changes in the closing prices of NASDAQ-100 companies prior to quarterly earnings reports. The task is to predict whether a stock's price will rise by more than 5% following the announcement. Labels are based on post-announcement movement, and the dataset includes a mix of positive and negative examples derived from real-world financial data.

## A.2 Few-shot Evaluation Datasets

| Name | # Samples | Task | Variables | Domain |
|---|---|---|---|---|
| MuJoCo | 3000 | Simulation | 28 | Physics |
| ETTm1 | 34369 | Forecasting | 7 | Electricity |
| ETTm2 | 34273 | Forecasting | 7 | Electricity |
| ETTh2 | 8353 | Forecasting | 7 | Electricity |
| Sine | 10000 | Synthetic Generation | 6 | Simulated |
| Weather | 36696 | Forecasting | 21 | Weather |
| ILI | 581 | Forecasting | 7 | Healthcare |
| SaugeenRiverFlow | 18921 | Forecasting | 1 | Weather |
| ECG200 | 200 | Classification | 1 | ECG |
| SelfRegulationSCP1 | 268 | Classification | 6 | EEG |
| StarLightCurves | 1000 | Classification | 1 | Sensor |
| AirQuality | 9357 | Generation | 13 | Nature |

Table 7: Summary of evaluation datasets used for few-shot generation, including number of samples, task type, number of variables, and domain. All datasets are framed as generation tasks.

We evaluate few-shot generalization on 12 datasets selected to cover diverse domains, temporal dynamics, and input dimensionalities. We treat all datasets as unconditional generative modeling tasks, even if they were originally used for tasks like classification or forecasting. A summary of all evaluation datasets, including the number of samples, task, number of channels, and domain type, is provided in Table 7.

**MuJoCo.** The Multi-Joint dynamics with Contact (MuJoCo) dataset [66] consists of simulated physical trajectories in robotic environments. Each sequence captures multivariate joint positions, velocities, and contact forces. It is commonly used in control and reinforcement learning settings.

**ETTm1, ETTm2, ETTh2.** The Electricity Transformer Temperature (ETT) datasets [75] contain load and oil temperature readings collected from electricity transformers at regular intervals. ETTm1 and ETTm2 are sampled at the minute level, while ETTh2 is sampled hourly. These datasets exhibit strong periodicity and seasonal patterns.

**Sine.** The Sine dataset is a synthetic multivariate time series dataset, where each sample consists of six channels indexed by $i$. Each time series is generated according to $x_t^{(i)}(j) = \sin(\eta t + \theta)$, where the frequency $\eta$ and phase $\theta$ are independently sampled from a uniform distribution: $\eta, \theta \sim \mathcal{U}[0, 0.1]$. Both $\eta$ and $\theta$ vary across samples and channels, resulting in diverse temporal patterns.

**Weather.** The Weather dataset [69] includes 21 meteorological variables recorded every 10 minutes over the full year of 2020. Features include air temperature, humidity, wind speed, and other environmental indicators.

**ILI.** The Influenza-Like Illness (ILI) dataset [12] contains weekly records of patient visits for ILI symptoms, reported by the CDC in the United States from 2002 to 2021. Each value reflects the ratio of ILI cases to total patient visits.

**Saugeen River Flow.**    The Saugeen dataset [47] is a long univariate time series recording the daily average river flow (in $m^3$/s) at Walkerton, Canada, from 1915 to 1979.

**ECG200.**    The ECG200 dataset [53] contains univariate electrocardiogram signals representing one heartbeat per sequence, labeled as normal or abnormal. It is a standard benchmark for time series classification.

**SelfRegulationSCP1.**    The SelfRegulationSCP1 dataset [2] includes multichannel EEG signals recorded during a Brain–Computer Interface (BCI) task. The subject was instructed to self-regulate slow cortical potentials (SCPs) to move a cursor. Each trial consists of 3.5-second EEG segments sampled at 256 Hz across six electrodes.

**StarLightCurves.**    The StarLightCurves dataset [56] contains univariate astronomical light curves of fixed length 1024. Each curve represents the brightness of a celestial object over time and is labeled into one of three variability-based categories.

**AirQuality.**    The AirQuality dataset [72] contains hourly averaged readings from five metal oxide chemical sensors integrated into an air quality chemical multisensor device. The device was deployed at road level in a heavily polluted area of an Italian city. Data were collected continuously from March 2004 to February 2005, resulting in the longest freely available on-field record of chemical sensor responses related to air quality.

# B Experimental Setting

This section provides detailed information about our model architecture, training procedures, hyper-parameters, and evaluation methodology. We split our discussion into pre-training and fine-tuning phases to reflect the two-stage setup used in all experiments.

## B.1 Model Architecture

Our generative model follows a denoising diffusion framework built on image-based architectures, specifically extending the Elucidated Diffusion Model (EDM) [36] and ImagenTime [50]. The model maps multivariate time series into image-like tensors via delay embedding, enabling reuse of visual backbone structures such as UNet. It incorporates two key additions: (1) a Dynamic Channel Adaptation module (DyConv) for handling varying channel counts across datasets, and (2) a dataset token for domain-specific conditioning. These innovations make our model suitable for unified pre-training across diverse domains.

**Delay embedding.** Following [50], given an input time series $\mathbf{x} \in \mathbb{R}^{L \times K}$ with $K$ channels and length $L$, we convert each univariate channel into a local trajectory matrix using delay embedding. For a skip parameter $m$ and column window size $n$, the transformation constructs a matrix:

$$
X = \begin{bmatrix} x_1 & x_{m+1} & \cdots & x_{L-n} \\ x_2 & x_{m+2} & \cdots & x_{L-n+1} \\ \vdots & \vdots & \ddots & \vdots \\ x_n & x_{m+n} & \cdots & x_L \end{bmatrix} \in \mathbb{R}^{n \times q} \,,
$$

where $q = \lceil \frac{L-n}{m} \rceil$. Each of the $K$ channels is processed this way and stacked into a tensor $x_{\text{img}} \in \mathbb{R}^{K \times n \times q}$, which is then zero-padded to a square shape $\mathbb{R}^{K \times n \times n}$ to fit the image backbone.

**UNet backbone.** The image $x_{\text{img}}$ is passed through a 2D UNet, similar to those used in diffusion models for vision. The architecture features residual blocks, downsampling and upsampling paths, and additional attention layers applied at specific resolutions. Temporal information is encoded using a noise-level embedding based on the EDM [36] design, and optional dataset-level context is injected via adaptive normalization layers.

**Dynamic channel adaptation (DyConv).** To support variable input/output channel dimensions across datasets, we introduce DyConv—a dynamic 2D convolution module. DyConv defines a canonical learnable convolutional kernel $W \in \mathbb{R}^{K \times K \times C_0 \times C_1}$, where $K = 3$ is the spatial kernel size and $C_0 = C_1 = 128$ are fixed reference channel dimensions. At runtime, the kernel is resized via bicubic interpolation over the channel dimensions to produce $W_{\text{interp}} \in \mathbb{R}^{K \times K \times C_{\text{in}} \times C_{\text{out}}}$, matching the actual dataset. The same interpolation applies to the bias. DyConv is used at the first and last layers of the UNet to map between dataset-specific inputs/outputs and a shared latent channel space. This mechanism allows a single model to operate on datasets with widely varying dimensionality without re-training.

**Dataset tokens.** To enable domain-specific conditioning, each dataset is assigned a unique learnable token. During training and generation, the token is embedded and injected into the denoising model via AdaGN layers [20], modulating intermediate activations based on dataset identity. This mechanism improves sample fidelity and enables multi-domain generation from a single model (see Sec.6.5). When evaluating without dataset tokens (unconditional setting), the model operates without this guidance.

**Dynamic Masking Implementation.** To support variable-length inputs within a fixed-size image representation, we generate a binary mask that identifies valid (non-padded) regions of each input. Specifically, after applying delay embedding and zero-padding to a fixed spatial size, we apply the same transformation to an all-ones tensor, producing a mask that aligns with the padded image. This mask dynamically adapts to the sequence length of each sample, while the overall image size remains fixed. During training, noise is added to the entire image, but the loss is computed only over the valid pixels, ensuring that the model focuses on meaningful regions and ignores padded areas.

## B.2 Training Procedure

We detail the procedure for each stage of our two-step framework and provide the complete pseudocode for both the pre-training and few-shot phases in Algorithm 1.

---

**Algorithm 1:** Unified diffusion pre-training and few-shot adaptation

---

**Input:** $\{\mathcal{D}^{(m)}\}_{m=1}^{M}$: heterogeneous datasets; $\mathcal{T}$: time-to-image transform; $N_\theta$: diffusion model; Tok: dataset token table; $\Sigma$: noise schedule; $\eta$: learning rate

**Output:** Learned parameters $\theta$

---

1: **Multi-domain Pre-training**;
2: **foreach** *training step* **do**
3:   $\mathcal{B} \leftarrow$ sample_batch$(\{\mathcal{D}^{(m)}\}_{m=1}^{M})$;
   $\triangleright$ Sample batch of series and dataset indices
4:   **foreach** $(\mathbf{x}, m) \in \mathcal{B}$ **do**
5:    $\mathbf{y} \leftarrow \text{Tok}[m]$ ;       $\triangleright$ Retrieve token for dataset $m$
6:    $x_{\text{img}}^0 \leftarrow$ pad_square$(\mathcal{T}(\mathbf{x}))$ ;     $\triangleright$ Transform and pad to image
7:    $\sigma \leftarrow$ sample$(\Sigma), \varepsilon \sim \mathcal{N}(0, \sigma^2 I)$ ;    $\triangleright$ Sample noise level
8:    $x_{\text{img}}^t \leftarrow x_{\text{img}}^0 + \varepsilon$ ;       $\triangleright$ Apply noise
9:    $\hat{\mathbf{x}} \leftarrow N_\theta(x_{\text{img}}^t, \sigma, \mathbf{y})$ ;      $\triangleright$ Predict clean image
10:    $\mathcal{L} \leftarrow \lambda(\sigma)\|\hat{\mathbf{x}} - x_{\text{img}}^0\|_2^2$ ;    $\triangleright$ Compute preconditioned loss
11:    $(\theta, \text{Tok}) \leftarrow$ AdamW$(\theta, \text{Tok}, \nabla\mathcal{L}, \eta)$ ;    $\triangleright$ Parameter update
12: **Few-shot Adaptation**;
13: Initialize new token $\mathbf{y}^*$ for $\mathcal{D}_{\text{new}}$ ;     $\triangleright$ Allocate for new domain
14: **foreach** *fine-tuning step* **do**
15:   $\mathcal{B} \leftarrow$ sample_batch$(\mathcal{D}_{\text{new}})$ ;      $\triangleright$ Sample few-shot batch
16:   **foreach** $\mathbf{x} \in \mathcal{B}$ **do**
17:    Repeat lines 5–11 with $\mathbf{y} \leftarrow \mathbf{y}^*$ ;   $\triangleright$ Use domain token for adaptation

---

### B.2.1 Pre-training Procedure

We pre-train our unified diffusion model across a diverse collection of time series datasets (see Table 6). This stage exposes the model to a wide spectrum of temporal dynamics, data modalities, and channel dimensionalities, encouraging it to develop transferable representations that generalize well under data scarcity. During pre-training, each sample is first converted into an image via delay embedding (see Section B.1), padded to a square resolution, and then diffused using Gaussian noise as defined in the Elucidated Diffusion Models (EDM) framework [36]. We apply our proposed **DyConv** layer to dynamically adapt to varying channel dimensions across datasets and condition the denoising network on a **dataset token**, allowing the model to incorporate domain-specific signals without requiring separate models per dataset. To improve stability and generalization, we maintain an exponential moving average (EMA) of the model weights throughout training. All training runs use two NVIDIA RTX 4090 GPUs and complete in approximately 4 hours. Following EDM preconditioning, noise levels are sampled from a fixed log-normal noise schedule. The full pre-training workflow, is described in Algorithm 1 (Multi-domain Pre-training). All hyperparameters specific to the pre-training phase are listed in Table 8, while the core architectural parameters, shared between both pre-training and fine-tuning, are summarized in Table 9.

### B.2.2 Training Loss

Our approach to defining the loss function for our diffusion process follows the methodologies presented in EDM [36] and ImagenTime [50]. More specifically, the objective of the model is to clean the noise via the following loss function:

$$\mathcal{L} = \mathbb{E}_{\sigma, x_{\text{img}}^0, \varepsilon}[\lambda(\sigma)\left\|\Gamma_\theta(x_{\text{img}}^\sigma; \sigma) - x_{\text{img}}^0\right\|_2^2] , \tag{4}$$

where $\lambda(\sigma)$ is a weighting function and $\Gamma_\theta$ is defined as follows:

$$\Gamma_\theta(x_{\text{img}}^\sigma; \sigma) = c_{\text{skip}}(\sigma)x_{\text{img}}^\sigma + c_{\text{out}}(\sigma)N_\theta(c_{\text{in}}(\sigma)x_{\text{img}}^\sigma; c_{\text{noise}}(\sigma); y) , \tag{5}$$

Table 8: Pre-training hyperparameters.

| Parameter | Value |
|---|---|
| Optimizer | AdamW |
| Learning rate | $1 \times 10^{-4}$ |
| Batch size | 2048 |
| Epochs | 1,000 |
| EMA decay | 0.9999 |
| Weight decay | $1 \times 10^{-5}$ |

Table 9: Model architecture hyperparameters used in both pre-training and fine-tuning.

| Component | Value |
|---|---|
| Delay embedding skip ($m$) | 8 |
| Delay embedding width ($n$) | 8 |
| Image resolution | $8 \times 8$ |
| UNet base channels | 32 |
| Channel multipliers | [1, 2, 2, 4] |
| Attention resolutions | [8, 4, 2] |
| Diffusion steps | 36 |
| DyConv kernel size | $3 \times 3$ |
| DyConv canonical shape | $[3, 3, 128, 128]$ |
| DyConv interpolation | Bicubic (channel dimensions) |

where $N_\theta$ is a neural network. $c_{\text{skip}}$ modulates the skip connection, $c_{\text{in}}$ and $c_{\text{out}}$ scale the input and output magnitudes, and $c_{\text{noise}}$ maps noise level $\sigma$ into a conditioning input for $N_\theta$. For robust training, we use the same values as in EDM's preconditioning [36], where the terms $c_{\text{skip}}(\sigma)$, $c_{\text{in}}(\sigma)$, $c_{\text{out}}(\sigma)$, and $c_{\text{noise}}(\sigma)$ are all functions of $\sigma$:

- $c_{\text{skip}}(\sigma) = \frac{\sigma_{\text{data}}^2}{\sigma^2 + \sigma_{\text{data}}^2}$

- $c_{\text{out}}(\sigma) = \frac{\sigma \cdot \sigma_{\text{data}}^2}{\sqrt{\sigma^2 + \sigma_{\text{data}}^2}}$

- $c_{\text{in}}(\sigma) = \frac{1}{\sqrt{\sigma^2 + \sigma_{\text{data}}^2}}$

- $c_{\text{noise}}(\sigma) = \frac{1}{4} \ln(\sigma)$

We use $\sigma_{\text{data}} = 0.5$ as a fixed parameter, and $\sigma(t)$ controls the noise level over time. These terms help stabilize the loss, which can vary with $\sigma$. Substituting Eq. 5 into Eq. 4 yields a per-sample loss weight. To balance this, EDM sets $\lambda(\sigma) = 1/c_{\text{out}}(\sigma)^2$, also ensuring uniform initial loss across the $\sigma$ range. Finally, we feed the network with an additional optional dataset token input $y$.

### B.2.3 Few-Shot Generation Adaptation

To adapt the pre-trained diffusion model to novel domains with limited supervision, we perform fine-tuning using a dedicated dataset token for each new dataset. For every unseen domain, we initialize a new token embedding that uniquely identifies the dataset (see Section B.1). This token is learned jointly with the model parameters during fine-tuning, guiding the generative process toward the target distribution while reusing the temporal representations acquired during pre-training. All architectural components, including delay embedding and DyConv, are retained during fine-tuning. Fine-tuning is performed for each small new dataset (see Table 7), in accordance with our few-shot benchmark protocol, and the same configuration is used across all datasets for consistency. The model is trained for 1,000 epochs. Note that due to the small size of this dataset, training is fast, and empirically we find that 100 epochs are sufficient for most cases. An exponential moving average (EMA) of the model weights is maintained throughout the fine-tuning phase to improve stability and sample quality. Due to the data-scarce nature of our few-shot settings, the effective batch size is set to the minimum of 2048 and the number of available samples in the dataset. All datasets are fine-tuned

using the same configuration. The overall adaptation procedure follows the workflow in Algorithm 1 (Few-shot Adaptation), with the new dataset token initialized and optimized during fine-tuning. All hyperparameters specific to the fine-tuning phase are listed in Table 10.

Table 10: Training hyperparameters used during the fine-tuning phase.

| Hyperparameter | Value |
| --- | --- |
| Optimizer | AdamW |
| Learning rate | $1 \times 10^{-4}$ |
| Epochs | 1,000 |
| Batch size | $\min(2048, \text{ # training samples})$ |
| EMA decay | 0.9999 |
| Weight decay | $1 \times 10^{-5}$ |

## B.3 Evaluation Protocol

We extend standardized time series generation metrics [73, 33] to broaden their applicability, allowing for consistent evaluation across both the original setup and few-shot learning setup. For each target dataset, we fine-tune the pre-trained model using either a small percentage of the training data (5%, 10%, 15%) or a fixed number of examples (10, 25, 50). In all cases, after fine-tuning, we evaluate the model by generating samples *in equal number to the original size of the test set* for each dataset. This ensures a fair comparison between generated and real data distributions and avoids bias from test-set size variability. We employ the following three metrics to evaluate different aspects of generative performance:

1. **Discriminative Score.** To quantitatively evaluate the similarity between real and generated time series data, we adopt the framework proposed by [73]. Specifically, we train a post-hoc LSTM-based time series classifier to distinguish between sequences originating from the original dataset (labeled as real) and those from the generated dataset (labeled as synthetic). The model is trained in a standard supervised learning setup, and the classification error on a held-out test set serves as a measure of distributional similarity. We then subtract this error from 0.5, such that a score of 0 indicates perfect indistinguishability, while higher scores reflect greater divergence.

2. **Predictive Score.** To assess the predictive utility of the generated data, we again follow an evaluation protocol proposed by [73], which tests whether synthetic data can support forecasting tasks. Specifically, we train an LSTM model on the synthetic dataset to perform next-step prediction: given a sequence of past time steps, the model forecasts the next temporal vector. The trained model is then evaluated on the original dataset using mean absolute error (MAE) as the performance metric. A low MAE indicates that the synthetic data captures the conditional temporal dynamics of the original data well enough to generalize to real sequences.

3. **Context-FID.** To measure global and contextual realism, we use the Context-Fréchet Inception Distance (Context-FID) [33]. This is an adaptation of the FID score used in image generation, but tailored for time series. Rather than using image-based features, Context-FID uses embeddings extracted from a contrastively trained encoder [23], separately trained for each dataset to capture temporal context. The final score reflects the Fréchet distance between the embedding distributions of real and synthetic samples. Lower values correspond to higher-quality generations.

# C  Additional Experiments

## C.1  Impact of Canonical Kernel Size in DyConv

To understand how the internal dimensionality of DyConv affects performance, we conduct an ablation study varying the size of the canonical kernel. All models share the same architecture and pre-training procedure (Sec. 5), differing only in the channel dimensions of DyConv's canonical kernel. Specifically, we vary the shape $[K, K, C_0, C_1]$ while keeping the kernel size fixed at $K = 3$. Each configuration is denoted DyConv$[C_0, C_1]$, and evaluated on the same few-shot benchmark. As shown in Table 11, The configurations DyConv[32,128] and DyConv[128,128] achieve similarly strong results, exhibiting low values in both discriminative score and contextFID. Interestingly, even the smaller DyConv[16,128] performs reasonably well, albeit with reduced effectiveness. In contrast, the smallest configuration, DyConv[1,128], shows a clear degradation across both metrics, highlighting the necessity of adequate parameterization. These findings support the conclusion that increasing the capacity of DyConv may yield better performance, with diminishing performance observed in extremely limited parameter regimes.

Table 11: Ablation study on the effect of DyConv's internal channel dimensions.

| Config | Disc. | c-FID | # Parameters |
|---|---|---|---|
| [1, 128] | 0.381 | 16.74 | 1,280 |
| [16, 128] | 0.149 | 1.945 | 18,560 |
| [32, 128] | **0.138** | 1.612 | 36,992 |
| [128, 128] | 0.143 | **1.451** | 147,584 |

## C.2  Effect of Fine-tuning Methods

We investigate two parameter-efficient fine-tuning (PEFT) strategies applied to our pre-trained diffusion model: Low-Rank Adaptation (LoRA) and Bias-Only tuning (BitFit). Both approaches aim to reduce the number of trainable parameters while enabling adaptation to new datasets in the few-shot regime. In the LoRA setup [31], we inject trainable low-rank matrices into the attention layers of our UNet architecture, specifically into the projection matrices $W_q$, $W_k$, $W_v$, and $W_o$ of each attention block, using a rank of $r = 16$. This configuration introduces only 126K trainable parameters. In the Bias-Only setup [3], all weights are frozen, and only the bias terms across all layers are updated, resulting in approximately 300K trainable parameters. As shown in Table 14, full fine-tuning consistently outperforms both LoRA and Bias-Only tuning across all subset sizes and all evaluation metrics, including contextFID, Discriminative Score, and Predictive Score. While LoRA and BitFit offer substantial parameter savings, their performance lags significantly behind full fine-tuning. This underscores the trade-off between efficiency and effectiveness in few-shot scenarios.

## C.3  Evaluating Sequence-Native Baselines Under the Few-Shot Setting

To further examine the role of the underlying backbone architecture, we conducted an additional experiment comparing our image-based framework to *sequence-native* generative models. Specifically, we considered DiffusionTS [74] and KoVAE [51], both of which operate directly in the temporal domain without converting sequences into images. Following the protocol of Section 6.1 , we pre-trained both baselines on the same heterogeneous corpus as our model and fine-tuned them under identical few-shot conditions. Table 12 reports the averaged Discriminative Score (Disc.), Predictive Score (Pred.), and contextFID (c-FID) across both percentage-based and fixed-count subset regimes. Across all subset sizes and evaluation metrics, our model consistently achieves the best results, demonstrating the effectiveness of the image-based backbone in few-shot regimes compared to sequence-native alternatives.

## C.4  Dataset Token Robustness Under Overlapping Domains

To evaluate the effectiveness of dataset token conditioning when domains are closely related, we pre-trained two variants of our model using only the ETT dataset family (ETTh1, ETTh2, ETTm1, ETTm2): one with dataset token conditioning and one without. Both models were evaluated on the

Table 12: Few-shot generation performance comparison between our unified model (pre-trained on sequence length 24) and sequence-native baselines. Lower is better for all metrics.

| Subset size | Metric | Ours (24) | DiffusionTS | KoVAE |
|---|---|---|---|---|
| 5% | Disc. | **0.110** | 0.244 | 0.331 |
| | Pred. | **0.458** | 0.471 | 0.500 |
| | c-FID | **0.674** | 2.668 | 3.719 |
| 10% | Disc. | **0.083** | 0.223 | 0.342 |
| | Pred. | **0.452** | 0.464 | 0.491 |
| | c-FID | **0.578** | 2.432 | 3.874 |
| 15% | Disc. | **0.066** | 0.223 | 0.338 |
| | Pred. | **0.451** | 0.466 | 0.486 |
| | c-FID | **1.086** | 2.475 | 3.549 |
| #10 | Disc. | **0.259** | 0.347 | 0.375 |
| | Pred. | **0.489** | 0.516 | 0.517 |
| | c-FID | **3.800** | 4.702 | 5.522 |
| #25 | Disc. | **0.190** | 0.310 | 0.352 |
| | Pred. | **0.467** | 0.492 | 0.499 |
| | c-FID | **1.582** | 3.817 | 3.844 |
| #50 | Disc. | **0.149** | 0.287 | 0.354 |
| | Pred. | **0.460** | 0.480 | 0.491 |
| | c-FID | **0.987** | 3.116 | 4.001 |

same ETT datasets without fine-tuning. This setup allows us to assess whether the dataset token can help disambiguate subtle inter-dataset variations during sampling. The results are summarized in Table 13.

Table 13: Evaluation of pre-trained models on the ETT dataset family with and without dataset token conditioning. Lower is better for all metrics.

| Dataset | Metric | w/ Dataset Token | w/o Dataset Token |
|---|---|---|---|
| ETTh1 | Disc. | **0.033** | 0.344 |
| | Pred. | **0.646** | 0.674 |
| | c-FID | **0.108** | 5.687 |
| ETTh2 | Disc. | **0.027** | 0.263 |
| | Pred. | **0.681** | 0.707 |
| | c-FID | **0.086** | 3.360 |
| ETTm1 | Disc. | **0.007** | 0.232 |
| | Pred. | **0.675** | 0.690 |
| | c-FID | **0.0210** | 2.920 |
| ETTm2 | Disc. | **0.007** | 0.236 |
| | Pred. | **0.694** | 0.718 |
| | c-FID | **0.0249** | 1.248 |

Across all four datasets, using the dataset token yields substantially better scores across all metrics compared to the model trained without it. This demonstrates that dataset token conditioning remains robust even in domains with overlapping or ambiguous boundaries, enabling the model to effectively distinguish subtle distributional differences during generation.

## C.5 Datasets Analysis

We investigate why our model fails to fine-tune effectively on certain datasets, such as `Weather` and `ECG200`, which remain challenging. We hypothesize that this difficulty stems from a failure to generalize to frequency distributions that were not encountered during pre-training.

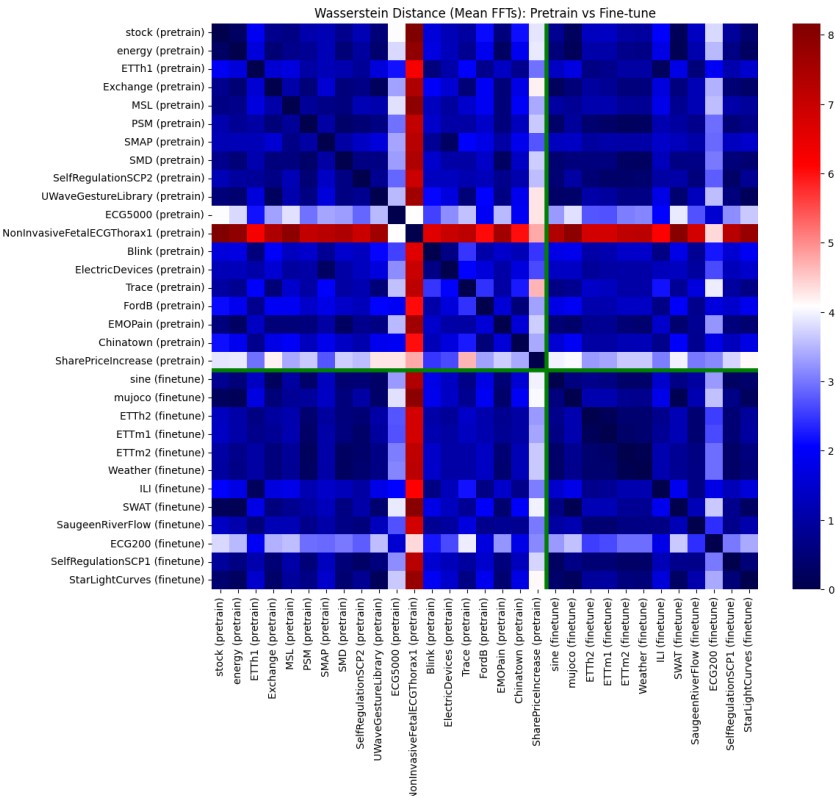

Figure 4: Wasserstein distance between dataset frequency distributions.

To test this hypothesis, we analyzed the frequency content of all datasets. We applied the **Fast Fourier Transform (FFT)** to every signal in both our pre-training and fine-tuning collections. For each dataset, we then computed a mean amplitude spectrum, $\alpha$, by averaging the amplitude vectors of all its signals:

$$\alpha = \frac{1}{N} \sum_{i=1}^{N} \alpha_i$$

where $N$ is the number of signals in the dataset and $\alpha_i$ is the amplitude spectrum vector for the $i$-th signal.

This vector $\alpha$ serves as a signature for the dataset's average frequency distribution. To quantify the dissimilarity in frequency content, we compute the **Wasserstein distance** between their respective normalized mean spectra. We denote this as $W(D, \tilde{D})$, where $D$ and $\tilde{D}$ are two different datasets from the collection of pre-trained and fine-tuned datasets. As shown in Fig. 4, our hypothesis is partially supported. The `ECG200` dataset, on which our model performs poorly, exhibits a large frequency distance relative to other datasets. However, `Weather`, another dataset where the model fails, shows a relatively small distance. This suggests that factors other than frequency distribution may be responsible for its low performance. Finally, while `NonInvasiveFetalECGThorax1` has the largest distance from all other datasets, this does not negatively affect performance, as it was included in the pre-training set.

### C.6 Full Results of Main Table

In this section, we present the complete results of our few-shot benchmark. Tables 15, 16, 17, 18, 19, 20 report the *Discriminative*, *Predictive*, and *contextFID* scores across all datasets (each table contains two datasets, together the tables represent all) and training subset sizes. Our model demonstrates marginal improvements over competing methods on several datasets, including **ETTh2**, **ETTm1**, **ILI**, and **SelfRegulationSCP1**, among others. Overall, it achieves the highest performance in 141

out of 168 cases in *Discriminative* and *contextFID* metrics combined. For brevity, we use acronyms for some datasets: **SRF** denotes *SaugeenRiverFlow*, **SCP1** denotes *SelfRegulationSCP1*, and **SLC** denotes *StarLightCurves*.

## C.7 Full results on pre-training datasets: Cond. vs. Uncond.

Table 21 provides the complete results on all pre-training datasets, comparing models trained with and without dataset token conditioning (Cond. vs. Uncond.). These evaluations were performed directly on the pre-training datasets without any further fine-tuning. The results demonstrate that conditioning with dataset tokens leads to consistently improved performance across most datasets and metrics, particularly in terms of contextFID and Discriminative Score. This reinforces the importance of explicit dataset-specific conditioning during pre-training for improving generation quality and alignment to the target distribution.

Table 14: Ablation study on fine-tuning methods. We compare LoRA (126K) and Bias-Only (300K) tuning across multiple subset sizes. Lower is better. Bold indicates best.

| Subset Size | Metric | Full FT (6M) | LoRA (126K) | Bias-Only (300K) |
|---|---|---|---|---|
| 5% | Disc. score | **0.110** | 0.317 | 0.340 |
| | Pred. score | **0.458** | 0.495 | 0.508 |
| | contextFID | **0.674** | 5.824 | 8.745 |
| 10% | Disc. score | **0.083** | 0.312 | 0.332 |
| | Pred. score | **0.452** | 0.469 | 0.489 |
| | contextFID | **0.578** | 5.297 | 8.397 |
| 15% | Disc. score | **0.066** | 0.302 | 0.337 |
| | Pred. score | **0.451** | 0.489 | 0.488 |
| | contextFID | **1.086** | 6.232 | 7.435 |
| #10 | Disc. score | **0.259** | 0.332 | 0.363 |
| | Pred. score | **0.489** | 0.496 | 0.544 |
| | contextFID | **3.800** | 7.396 | 9.417 |
| #25 | Disc. score | **0.190** | 0.323 | 0.347 |
| | Pred. score | **0.467** | 0.484 | 0.514 |
| | contextFID | **1.582** | 6.267 | 9.173 |
| #50 | Disc. score | **0.149** | 0.326 | 0.343 |
| | Pred. score | **0.460** | 0.494 | 0.517 |
| | contextFID | **0.987** | 6.345 | 9.882 |

Table 15: Main Table Results - Part 1. The table reports discriminative and predictive scores along with their standard deviations, as well as the contextFID score.

| dataset | Subset | Metric | Our[24] | ImagenTime | DiffTS | KoVAE | TimeGAN |
|---------|--------|--------|---------|------------|--------|-------|---------|
| AirQuality | 5% | Disc. | **.065±.04** | .49±.0 | .177±.01 | .458±.03 | .441±.04 |
| | | Pred. | .011±.0 | .019±.01 | .007±.0 | .043±.0 | **.003±.0** |
| | | c-FID | **0.441** | 6.858 | 0.721 | 3.971 | 1.317 |
| | 10% | Disc. | **.085±.07** | .451±.01 | .171±.01 | .428±.04 | .322±.12 |
| | | Pred. | .01±.0 | .009±.0 | .006±.0 | .041±.0 | **.005±.0** |
| | | c-FID | **0.419** | 1.476 | 0.657 | 3.336 | 2.995 |
| | 15% | Disc. | **.066±.06** | .46±.01 | .174±.01 | .422±.02 | .445±.04 |
| | | Pred. | .009±.0 | .009±.0 | .006±.0 | .04±.0 | **.002±.0** |
| | | c-FID | **0.421** | 1.65 | 0.583 | 3.571 | 2.089 |
| | 100% | Disc. | **.065±.03** | .087±.04 | .116±.0 | .333±.03 | .43±.07 |
| | | Pred. | .006±.0 | .009±.0 | .005±.0 | .039±.0 | **.004±.0** |
| | | c-FID | 0.284 | 0.401 | **0.282** | 2.589 | 2.566 |
| | 10# | Disc. | .374±.03 | .494±.01 | .406±.02 | .488±.0 | **.286±.12** |
| | | Pred. | .041±.0 | .044±.0 | .044±.0 | .044±.0 | **.002±.0** |
| | | c-FID | **1.359** | 4.43 | 5.713 | 3.994 | 2.415 |
| | 25# | Disc. | **.322±.18** | .426±.14 | .374±.12 | .483±.01 | .387±.09 |
| | | Pred. | .019±.0 | **.017±.0** | .038±.01 | .044±.0 | .023±.01 |
| | | c-FID | **1.099** | 3.184 | 3.363 | 4.221 | 2.89 |
| | 50# | Disc. | **.303±.15** | .406±.14 | .393±.04 | .483±.01 | .365±.12 |
| | | Pred. | .019±.0 | .014±.0 | .016±.0 | .044±.0 | **.004±.0** |
| | | c-FID | **0.841** | 2.706 | 2.599 | 3.433 | 1.603 |
| ECG200 | 5% | Disc. | **.098±.05** | .115±.06 | .292±.07 | .383±.04 | .463±.09 |
| | | Pred. | 1.061±.0 | 1.061±.0 | 1.061±.0 | 1.062±.0 | **.455±.0** |
| | | c-FID | 0.866 | **0.793** | 2.15 | 4.199 | 40.207 |
| | 10% | Disc. | .128±.08 | **.08±.05** | .328±.07 | .338±.06 | .385±.06 |
| | | Pred. | 1.061±.0 | 1.061±.0 | 1.061±.0 | 1.062±.0 | **.994±.0** |
| | | c-FID | 0.884 | **0.605** | 2.401 | 3.51 | 11.225 |
| | 15% | Disc. | **.037±.03** | .065±.05 | .32±.06 | .297±.08 | .398±.14 |
| | | Pred. | 1.062±.0 | 1.061±.0 | 1.06±.0 | 1.062±.0 | **.916±.0** |
| | | c-FID | **0.315** | 0.41 | 2.471 | 3.226 | 24.112 |
| | 100% | Disc. | **.037±.03** | .072±.04 | .347±.06 | .307±.14 | .39±.13 |
| | | Pred. | 1.062±.0 | 1.063±.0 | 1.062±.0 | 1.063±.0 | **.869±.0** |
| | | c-FID | 4.194 | 4.169 | 2.431 | **1.661** | 6.445 |
| | 10# | Disc. | **.117±.06** | **.117±.05** | .35±.07 | .35±.06 | .385±.06 |
| | | Pred. | 1.061±.0 | 1.061±.0 | 1.06±.0 | 1.062±.0 | **.994±.0** |
| | | c-FID | **0.496** | 0.805 | 2.859 | 3.983 | 11.225 |
| | 25# | Disc. | **.063±.05** | .072±.04 | .345±.07 | .32±.07 | .372±.12 |
| | | Pred. | 1.063±.0 | 1.062±.0 | 1.061±.0 | 1.063±.0 | **.972±.0** |
| | | c-FID | **0.148** | 0.232 | 2.03 | 2.338 | 9.289 |
| | 50# | Disc. | **.06±.02** | .063±.04 | .318±.1 | .27±.13 | .333±.15 |
| | | Pred. | 1.062±.0 | 1.062±.0 | 1.093±.0 | 1.062±.0 | **1.048±.0** |
| | | c-FID | **0.113** | 0.172 | 2.493 | 1.662 | 9.121 |

Table 16: Main Table Results - Part 2. The table reports discriminative and predictive scores along with their standard deviations, as well as the contextFID score.

| dataset | Subset | Metric | Our[24] | ImagenTime | DiffTS | KoVAE | TimeGAN |
|---|---|---|---|---|---|---|---|
| ETTh2 | 5% | Disc. | **.034±.01** | .296±.1 | .27±.02 | .474±.01 | .49±.0 |
| | | Pred. | **.688±.0** | .707±.0 | .742±.01 | .795±.01 | .921±.04 |
| | | c-FID | **0.175** | 0.691 | 2.112 | 5.322 | 13.861 |
| | 10% | Disc. | **.024±.01** | .257±.1 | .268±.02 | .461±.01 | .485±.01 |
| | | Pred. | **.688±.0** | .698±.0 | .72±.01 | .819±.01 | .834±.02 |
| | | c-FID | **0.148** | 0.704 | 1.653 | 4.621 | 17.031 |
| | 15% | Disc. | **.017±.01** | .296±.1 | .271±.01 | .458±.0 | .458±.01 |
| | | Pred. | **.684±.0** | .703±.0 | .716±.01 | .826±.0 | .725±.01 |
| | | c-FID | **0.122** | 0.582 | 1.709 | 4.912 | 7.399 |
| | 100% | Disc. | **.015±.01** | .025±.01 | .27±.03 | .47±.01 | .491±.0 |
| | | Pred. | **.676±.0** | .684±.0 | .703±.01 | .837±.01 | .929±.04 |
| | | c-FID | **0.07** | 0.145 | 1.613 | 4.739 | 15.177 |
| | 10# | Disc. | **.266±.04** | .432±.08 | .332±.06 | .495±.0 | .484±.02 |
| | | Pred. | **.717±.01** | .728±.0 | .751±.0 | .804±.01 | .836±.01 |
| | | c-FID | **1.718** | 3.173 | 3.662 | 5.996 | 62.157 |
| | 25# | Disc. | **.186±.02** | .389±.08 | .338±.04 | .495±.0 | .451±.04 |
| | | Pred. | **.703±.0** | .722±.0 | .747±.02 | .831±.01 | 1.208±.07 |
| | | c-FID | **1.113** | 1.821 | 2.455 | 5.417 | 16.998 |
| | 50# | Disc. | **.15±.02** | .44±.03 | .29±.05 | .495±.0 | .483±.02 |
| | | Pred. | **.705±.0** | .721±.0 | .752±.01 | .755±.01 | .907±.03 |
| | | c-FID | **0.565** | 1.404 | 2.143 | 5.825 | 35.516 |
| ETTm1 | 5% | Disc. | **.015±.01** | .386±.11 | .317±.02 | .47±.01 | .455±.02 |
| | | Pred. | **.681±.0** | .698±.0 | .699±.01 | .722±.01 | .924±.02 |
| | | c-FID | **0.042** | 1.036 | 1.9 | 4.823 | 4.671 |
| | 10% | Disc. | **.011±.01** | .071±.02 | .315±.02 | .473±.01 | .46±.01 |
| | | Pred. | **.68±.0** | .686±.0 | .691±.0 | .732±.01 | .892±.03 |
| | | c-FID | **0.044** | 0.223 | 1.937 | 4.33 | 7.64 |
| | 15% | Disc. | **.013±.0** | .034±.01 | .321±.01 | .466±.01 | .461±.03 |
| | | Pred. | .678±.0 | .681±.0 | .704±.0 | .725±.0 | **.526±.01** |
| | | c-FID | **0.029** | 0.097 | 1.937 | 4.461 | 10.099 |
| | 100% | Disc. | **.004±.0** | .01±.0 | .324±.02 | .458±.01 | .446±.05 |
| | | Pred. | **.675±.0** | **.675±.0** | .707±.01 | .723±.01 | .775±.03 |
| | | c-FID | **0.011** | 0.025 | 1.867 | 4.876 | 4.543 |
| | 10# | Disc. | **.342±.03** | .449±.03 | .441±.01 | .492±.0 | .46±.03 |
| | | Pred. | .847±.02 | .778±.0 | .912±.01 | .752±.01 | **.708±.01** |
| | | c-FID | **4.325** | 4.944 | 4.731 | 6.749 | 7.268 |
| | 25# | Disc. | **.216±.02** | .496±.0 | .378±.03 | .476±.01 | .463±.03 |
| | | Pred. | .742±.01 | .708±.0 | .903±.02 | .734±.01 | **.701±.02** |
| | | c-FID | **1.623** | 5.115 | 2.929 | 4.695 | 5.186 |
| | 50# | Disc. | **.148±.01** | .493±.0 | .384±.02 | .471±.01 | .464±.02 |
| | | Pred. | .694±.0 | .703±.0 | .833±.02 | .718±.0 | **.691±.01** |
| | | c-FID | **0.651** | 3.77 | 2.317 | 4.943 | 11.111 |

Table 17: Main Table Results - Part 3. The table reports discriminative and predictive scores along with their standard deviations, as well as the contextFID score.

| dataset | Subset | Metric | Our[24] | ImagenTime | DiffTS | KoVAE | TimeGAN |
|---|---|---|---|---|---|---|---|
| ETTm2 | 5% | Disc. | **.021±.01** | .444±.01 | .241±.02 | .452±.01 | .4±.08 |
| | | Pred. | **.7±.0** | .727±.0 | .736±.01 | .793±.01 | .706±.02 |
| | | c-FID | **0.071** | 0.995 | 1.593 | 5.077 | 5.184 |
| | 10% | Disc. | **.011±.01** | .065±.02 | .234±.01 | .437±.01 | .482±.01 |
| | | Pred. | **.697±.0** | .71±.0 | .72±.01 | .808±.02 | 1.106±.08 |
| | | c-FID | **0.044** | 0.346 | 1.541 | 4.738 | 6.915 |
| | 15% | Disc. | **.013±.0** | .041±.01 | .233±.02 | .407±.02 | .475±.01 |
| | | Pred. | .7±.0 | .705±.0 | .725±.01 | .753±.0 | **.677±.02** |
| | | c-FID | **0.06** | 0.203 | 1.529 | 3.528 | 10.711 |
| | 100% | Disc. | **.004±.0** | .011±.0 | .228±.03 | .378±.04 | .478±.01 |
| | | Pred. | **.693±.0** | .694±.0 | .713±.0 | .751±.0 | .867±.01 |
| | | c-FID | **0.016** | 0.038 | 1.701 | 3.643 | 7.742 |
| | 10# | Disc. | **.343±.01** | .371±.01 | .378±.02 | .487±.01 | .495±.0 |
| | | Pred. | **.771±.01** | .796±.02 | .812±.02 | .806±.01 | .875±.06 |
| | | c-FID | **3.679** | 4.72 | 4.015 | 9.069 | 179.951 |
| | 25# | Disc. | **.211±.02** | .487±.0 | .346±.06 | .479±.01 | .394±.04 |
| | | Pred. | **.731±.01** | .773±.0 | .75±.01 | .785±.01 | .831±.01 |
| | | c-FID | **1.644** | 4.257 | 2.836 | 5.458 | 7.136 |
| | 50# | Disc. | **.135±.02** | .472±.01 | .276±.02 | .475±.01 | .455±.04 |
| | | Pred. | .721±.0 | .749±.0 | .768±.01 | .793±.02 | **.662±.02** |
| | | c-FID | **0.981** | 2.527 | 2.469 | 5.776 | 11.327 |
| ILI | 5% | Disc. | **.194±.04** | .483±.02 | .393±.04 | .489±.01 | .445±.03 |
| | | Pred. | .563±.0 | .582±.0 | .571±.0 | .61±.01 | **.556±.02** |
| | | c-FID | **1.135** | 5.51 | 3.274 | 5.828 | 8.697 |
| | 10% | Disc. | **.113±.04** | .459±.05 | .399±.02 | .493±.0 | .439±.02 |
| | | Pred. | **.555±.0** | .575±.0 | .576±.01 | .605±.01 | .57±.03 |
| | | c-FID | **0.489** | 3.29 | 3.286 | 6.098 | 4.965 |
| | 15% | Disc. | **.077±.04** | .447±.04 | .403±.03 | .49±.01 | .389±.08 |
| | | Pred. | **.554±.0** | .572±.0 | .564±.01 | .606±.01 | .581±.01 |
| | | c-FID | **0.416** | 2.56 | 2.805 | 6.039 | 4.594 |
| | 100% | Disc. | **.064±.03** | .388±.12 | .368±.02 | .489±.01 | .407±.03 |
| | | Pred. | .554±.0 | .573±.0 | **.552±.0** | .605±.0 | .617±.02 |
| | | c-FID | **0.329** | 2.229 | 1.989 | 7.5 | 4.15 |
| | 10# | Disc. | **.29±.05** | .491±.01 | .435±.02 | .493±.01 | .458±.04 |
| | | Pred. | .573±.01 | .628±.01 | .66±.02 | .607±.01 | **.453±.01** |
| | | c-FID | **2.115** | 7.82 | 5.406 | 6.576 | 6.275 |
| | 25# | Disc. | **.194±.04** | .489±.01 | .425±.03 | .495±.0 | .448±.03 |
| | | Pred. | .567±.01 | .586±.0 | .585±.01 | .628±.01 | **.536±.01** |
| | | c-FID | **0.996** | 5.939 | 4.085 | 6.809 | 8.127 |
| | 50# | Disc. | **.121±.03** | .479±.01 | .376±.05 | .493±.01 | .431±.02 |
| | | Pred. | **.555±.0** | .58±.0 | .574±.01 | .62±.01 | .62±.0 |
| | | c-FID | **0.615** | 4.102 | 3.081 | 6.093 | 5.333 |

Table 18: Main Table Results - Part 4. The table reports discriminative and predictive scores along with their standard deviations, as well as the contextFID score. **SRF** denotes *SaugeenRiverFlow* and **SCP1** denotes *SelfRegulationSCP1*.

| dataset | Subset | Metric | Our[24] | ImagenTime | DiffTS | KoVAE | TimeGAN |
|---------|--------|--------|---------|------------|--------|-------|---------|
| SRF | 5% | Disc. | **.002±.0** | .011±.01 | .147±.02 | .137±.09 | .08±.06 |
| | | Pred. | .604±.0 | .605±.0 | **.602±.0** | .605±.0 | .744±.0 |
| | | c-FID | **0.016** | 0.102 | 1.134 | 2.44 | 2.416 |
| | 10% | Disc. | **.004±.0** | .008±.0 | .13±.01 | .122±.08 | .354±.07 |
| | | Pred. | .604±.0 | .604±.0 | .602±.0 | .605±.0 | **.038±.0** |
| | | c-FID | **0.012** | 0.066 | 1.154 | 2.211 | 78.061 |
| | 15% | Disc. | **.006±.0** | .007±.0 | .123±.01 | .109±.04 | .095±.06 |
| | | Pred. | .605±.0 | .604±.0 | **.603±.0** | .604±.0 | .762±.0 |
| | | c-FID | 0.038 | **0.033** | 1.087 | 1.494 | 0.961 |
| | 100% | Disc. | .004±.0 | **.003±.0** | .111±.01 | .094±.05 | .067±.03 |
| | | Pred. | .605±.0 | .605±.0 | **.603±.0** | .603±.0 | .726±.0 |
| | | c-FID | 0.007 | **0.005** | 0.988 | 1.531 | 4.384 |
| | 10# | Disc. | **.051±.02** | .064±.03 | .368±.14 | .24±.01 | .14±.07 |
| | | Pred. | .605±.0 | .604±.0 | .602±.0 | .601±.0 | **.473±.0** |
| | | c-FID | 0.815 | **0.706** | 6.409 | 2.732 | 22.553 |
| | 25# | Disc. | .045±.02 | **.02±.02** | .304±.13 | .202±.06 | .117±.06 |
| | | Pred. | .603±.0 | .604±.0 | .603±.0 | .606±.0 | **.474±.0** |
| | | c-FID | 0.417 | **0.308** | 7.291 | 3.254 | 3.111 |
| | 50# | Disc. | **.013±.01** | .015±.01 | .287±.12 | .187±.11 | .16±.08 |
| | | Pred. | .605±.0 | .604±.0 | .602±.0 | .604±.0 | **.402±.0** |
| | | c-FID | **0.179** | 0.233 | 7.597 | 3.519 | 2.612 |
| SCP1 | 5% | Disc. | **.239±.06** | .332±.06 | .347±.04 | .452±.02 | .433±.07 |
| | | Pred. | .468±.01 | .491±.02 | .564±.04 | **.451±.01** | .624±.03 |
| | | c-FID | **1.159** | 2.97 | 3.671 | 8.329 | 8.149 |
| | 10% | Disc. | **.083±.05** | .312±.15 | .275±.05 | .409±.03 | .428±.03 |
| | | Pred. | .437±.0 | **.431±.0** | .534±.02 | .462±.01 | .52±.02 |
| | | c-FID | **0.959** | 2.237 | 2.912 | 8.69 | 5.403 |
| | 15% | Disc. | **.067±.04** | .22±.1 | .274±.03 | .405±.03 | .418±.03 |
| | | Pred. | .433±.0 | **.433±.0** | .472±.02 | .469±.01 | .558±.02 |
| | | c-FID | **0.575** | 2.821 | 2.546 | 8.36 | 4.99 |
| | 100% | Disc. | **.061±.04** | .284±.14 | .123±.03 | .382±.04 | .399±.04 |
| | | Pred. | .424±.0 | .431±.0 | .438±.01 | .439±.01 | **.415±.01** |
| | | c-FID | **0.467** | 1.593 | 1.372 | 6.34 | 3.75 |
| | 10# | Disc. | **.269±.04** | .364±.05 | .411±.03 | .434±.02 | .407±.07 |
| | | Pred. | .46±.01 | .462±.01 | .568±.05 | .466±.01 | **.343±.02** |
| | | c-FID | **1.293** | 2.958 | 3.433 | 6.792 | 4.702 |
| | 25# | Disc. | **.142±.04** | .426±.07 | .304±.04 | .396±.02 | .448±.04 |
| | | Pred. | .439±.0 | **.431±.0** | .487±.02 | .459±.01 | .702±.02 |
| | | c-FID | **0.808** | 2.582 | 2.304 | 7.037 | 5.965 |
| | 50# | Disc. | **.052±.03** | .404±.09 | .281±.04 | .427±.02 | .36±.08 |
| | | Pred. | .433±.0 | **.432±.0** | .467±.01 | .461±.01 | .434±.01 |
| | | c-FID | **0.694** | 3.042 | 2.726 | 9.436 | 5.766 |

Table 19: Main Table Results - Part 5. The table reports discriminative and predictive scores along with their standard deviations, as well as the contextFID score. **SLC** denotes *StarLightCurves*.

| dataset | Subset | Metric | Our[24] | ImagenTime | DiffTS | KoVAE | TimeGAN |
|---------|--------|--------|---------|------------|--------|-------|---------|
| SLC | 5% | Disc. | .042±.03 | **.026±.02** | .096±.03 | .225±.09 | .04±.02 |
| | | Pred. | .518±.0 | .501±.0 | .545±.0 | .502±.0 | **.474±.0** |
| | | c-FID | 0.201 | **0.053** | 0.406 | 3.147 | 0.295 |
| | 10% | Disc. | **.013±.01** | .017±.01 | .07±.03 | .199±.07 | .058±.03 |
| | | Pred. | .497±.0 | .498±.0 | .514±.0 | .504±.0 | **.407±.0** |
| | | c-FID | **0.01** | 0.037 | 0.224 | 2.902 | 0.234 |
| | 15% | Disc. | **.013±.01** | .017±.01 | .056±.03 | .192±.07 | .08±.03 |
| | | Pred. | **.498±.0** | .5±.0 | .508±.0 | .506±.0 | .592±.0 |
| | | c-FID | **0.008** | 0.056 | 0.128 | 3.608 | 0.248 |
| | 100% | Disc. | .021±.02 | **.018±.01** | .036±.02 | .24±.08 | .047±.02 |
| | | Pred. | **.497±.0** | .498±.0 | .501±.0 | .508±.0 | .623±.0 |
| | | c-FID | 0.009 | **0.005** | 0.096 | 1.997 | 0.302 |
| | 10# | Disc. | **.018±.02** | .066±.04 | .043±.03 | .31±.04 | .133±.05 |
| | | Pred. | .498±.0 | .526±.0 | **.497±.0** | .55±.0 | .519±.0 |
| | | c-FID | **0.094** | 0.26 | 0.262 | 4.833 | 1.865 |
| | 25# | Disc. | .046±.04 | **.026±.01** | .065±.04 | .263±.08 | .06±.03 |
| | | Pred. | .515±.0 | **.498±.0** | **.498±.0** | .519±.0 | .591±.0 |
| | | c-FID | 0.346 | **0.04** | 0.54 | 3.338 | 0.176 |
| | 50# | Disc. | .018±.02 | **.017±.01** | .14±.06 | .226±.07 | .04±.02 |
| | | Pred. | .503±.0 | .497±.0 | .576±.0 | .502±.0 | **.474±.0** |
| | | c-FID | 0.066 | **0.048** | 0.438 | 2.968 | 0.295 |
| Weather | 5% | Disc. | .479±.01 | .499±.0 | **.416±.01** | .499±.0 | .5±.0 |
| | | Pred. | **.061±.0** | .079±.0 | .07±.0 | .08±.0 | .231±.01 |
| | | c-FID | **3.679** | 9.092 | 7.425 | 12.823 | 39.294 |
| | 10% | Disc. | .433±.01 | .498±.0 | **.421±.0** | .498±.0 | .5±.0 |
| | | Pred. | **.06±.0** | .07±.0 | .07±.0 | .104±.01 | .226±.0 |
| | | c-FID | **3.631** | 13.005 | 7.19 | 17.045 | 30.8 |
| | 15% | Disc. | **.41±.03** | .495±.0 | .417±.01 | .498±.0 | .494±.01 |
| | | Pred. | **.059±.0** | .067±.0 | .069±.0 | .08±.0 | .157±.0 |
| | | c-FID | 10.86 | **7.214** | 9.683 | 22.476 | 29.475 |
| | 100% | Disc. | **.034±.01** | .204±.06 | .415±.0 | .499±.0 | .498±.0 |
| | | Pred. | **.055±.0** | .058±.0 | .07±.0 | .087±.01 | .081±.0 |
| | | c-FID | **4.97** | 7.2 | 8.871 | 23.638 | 41.502 |
| | 10# | Disc. | **.483±.01** | .499±.0 | .492±.01 | .5±.0 | .494±.0 |
| | | Pred. | .087±.0 | .093±.01 | .103±.01 | .246±.01 | **.022±.0** |
| | | c-FID | 26.481 | 15.305 | 15.016 | **14.954** | 35.694 |
| | 25# | Disc. | .499±.0 | .5±.0 | **.482±.01** | .5±.0 | .499±.0 |
| | | Pred. | .068±.0 | .07±.0 | .082±.0 | .327±.03 | **.042±.01** |
| | | c-FID | **8.1** | 13.339 | 12.78 | 20.065 | 22.921 |
| | 50# | Disc. | .497±.0 | .499±.0 | **.463±.01** | .499±.0 | .49±.02 |
| | | Pred. | .064±.0 | .07±.0 | .076±.0 | .277±.04 | **.026±.0** |
| | | c-FID | **5.999** | 10.353 | 14.377 | 18.868 | 23.114 |

Table 20: Main Table Results - Part 6. The table reports discriminative and predictive scores along with their standard deviations, as well as the contextFID score.

| dataset | Subset | Metric | Our[24] | ImagenTime | DiffTS | KoVAE | TimeGAN |
|---------|--------|--------|---------|------------|--------|-------|---------|
| mujoco | 5% | Disc. | **.111±.02** | .499±.0 | .213±.03 | .47±.02 | .416±.02 |
| | | Pred. | **.04±.0** | .065±.0 | .044±.0 | .062±.0 | .083±.0 |
| | | c-FID | **0.249** | 11.233 | 0.556 | 5.231 | 1.638 |
| | 10% | Disc. | **.074±.02** | .499±.0 | .142±.02 | .447±.03 | .412±.06 |
| | | Pred. | **.04±.0** | .062±.0 | **.04±.0** | .052±.0 | .073±.0 |
| | | c-FID | **0.248** | 8.841 | 0.312 | 4.598 | 1.3 |
| | 15% | Disc. | **.059±.03** | .497±.0 | .121±.02 | .434±.03 | .408±.05 |
| | | Pred. | **.038±.0** | .063±.0 | .039±.0 | .046±.0 | .072±.01 |
| | | c-FID | **0.163** | 8.674 | 0.252 | 4.307 | 1.372 |
| | 100% | Disc. | **.008±.01** | .25±.07 | .086±.01 | .455±.01 | .411±.1 |
| | | Pred. | **.033±.0** | .042±.0 | .036±.0 | .045±.0 | .077±.0 |
| | | c-FID | **0.031** | 0.85 | 0.16 | 4.078 | 1.552 |
| | 10# | Disc. | .38±.02 | .499±.0 | .379±.1 | .496±.0 | **.338±.09** |
| | | Pred. | .09±.0 | **.066±.0** | .142±.02 | .132±.01 | .109±.0 |
| | | c-FID | 2.25 | 13.939 | 4.263 | 5.856 | **2.023** |
| | 25# | Disc. | **.294±.03** | .5±.0 | .38±.02 | .491±.01 | .371±.04 |
| | | Pred. | .063±.0 | .06±.0 | **.052±.0** | .087±.0 | .106±.0 |
| | | c-FID | 2.076 | 14.019 | 3.524 | 6.4 | **1.891** |
| | 50# | Disc. | **.204±.02** | .499±.0 | .3±.03 | .491±.0 | .327±.14 |
| | | Pred. | .054±.0 | .065±.0 | **.048±.0** | .093±.0 | .129±.0 |
| | | c-FID | **0.848** | 11.878 | 2.051 | 5.419 | 1.251 |
| sine | 5% | Disc. | **.019±.01** | .267±.07 | .07±.02 | .162±.04 | .492±.0 |
| | | Pred. | .096±.0 | .096±.0 | .102±.0 | **.095±.0** | .264±.0 |
| | | c-FID | **0.051** | 2.232 | 0.105 | 1.597 | 6.93 |
| | 10% | Disc. | **.014±.01** | .257±.1 | .039±.01 | .199±.06 | .494±.0 |
| | | Pred. | **.094±.0** | .095±.0 | .098±.0 | .095±.0 | .267±.01 |
| | | c-FID | **0.048** | 2.257 | 0.056 | 1.827 | 2.12 |
| | 15% | Disc. | **.014±.01** | .257±.11 | .041±.01 | .181±.05 | .495±.0 |
| | | Pred. | **.094±.0** | .096±.0 | .098±.0 | **.094±.0** | .269±.01 |
| | | c-FID | **0.025** | 2.233 | 0.038 | 1.746 | 3.431 |
| | 100% | Disc. | .009±.01 | **.008±.01** | .018±.01 | .173±.04 | .495±.0 |
| | | Pred. | **.094±.0** | **.094±.0** | .097±.0 | **.094±.0** | .263±.01 |
| | | c-FID | **0.005** | 0.019 | 0.015 | 1.619 | 2.843 |
| | 10# | Disc. | **.18±.02** | .437±.04 | .311±.01 | .336±.04 | .499±.0 |
| | | Pred. | .12±.0 | .119±.0 | .144±.0 | **.098±.0** | .266±.01 |
| | | c-FID | **0.974** | 5.662 | 4.035 | 2.457 | 6.733 |
| | 25# | Disc. | **.068±.04** | .37±.1 | .314±.03 | .284±.05 | .5±.0 |
| | | Pred. | .098±.0 | **.097±.0** | .172±.05 | .1±.0 | .267±.01 |
| | | c-FID | **0.614** | 4.356 | 3.621 | 2.173 | 3.636 |
| | 50# | Disc. | **.087±.01** | .312±.09 | .246±.01 | .25±.08 | .499±.0 |
| | | Pred. | .099±.0 | .097±.0 | .113±.0 | **.096±.0** | .266±.01 |
| | | c-FID | **0.292** | 3.429 | 1.222 | 1.851 | 9.881 |

Table 21: Full results on pre-training datasets: conditional (with dataset token) vs. unconditional (without dataset token). Results are reported without any additional fine-tuning on the individual datasets.

| Dataset | Metric | Cond. | Uncond. |
|---|---|---|---|
| Blink | contextFID | **0.320** | 0.268 |
| | Disc score | **0.028** | 0.050 |
| Chinatown | contextFID | **0.157** | 27.923 |
| | Disc score | 0.188 | **0.175** |
| ECG5000 | contextFID | **0.033** | 4.920 |
| | Disc score | **0.016** | 0.176 |
| EMOPain | contextFID | 5.877 | **5.433** |
| | Disc score | **0.491** | 0.492 |
| ETTh1 | contextFID | **0.474** | 1.309 |
| | Disc score | **0.159** | 0.232 |
| ElectricDevices | contextFID | **0.097** | 2.677 |
| | Disc score | **0.009** | 0.119 |
| Exchange | contextFID | **0.160** | 0.353 |
| | Disc score | **0.014** | 0.096 |
| FordB | contextFID | **0.017** | 2.448 |
| | Disc score | **0.008** | 0.093 |
| MSL | contextFID | **28.504** | 29.708 |
| | Disc score | **0.500** | **0.500** |
| NonInvasiveFetalECGThorax1 | contextFID | **0.009** | 5.701 |
| | Disc score | **0.013** | 0.427 |
| PSM | contextFID | **1.468** | 8.195 |
| | Disc score | 0.493 | **0.267** |
| SMAP | contextFID | **22.069** | 29.311 |
| | Disc score | 0.499 | **0.423** |
| SMD | contextFID | **17.111** | 22.482 |
| | Disc score | **0.499** | **0.499** |
| SelfRegulationSCP2 | contextFID | **0.877** | 10.772 |
| | Disc score | **0.058** | 0.268 |
| SharePriceIncrease | contextFID | **0.145** | 2.070 |
| | Disc score | **0.019** | 0.163 |
| Trace | contextFID | **0.044** | 48.133 |
| | Disc score | **0.083** | 0.215 |
| UWaveGestureLibrary | contextFID | **0.118** | 0.304 |
| | Disc score | 0.071 | **0.052** |
| energy | contextFID | 14.291 | **8.080** |
| | Disc score | **0.500** | **0.500** |
| stock | contextFID | **0.162** | 0.244 |
| | Disc score | **0.034** | 0.051 |

