# OpenReview forum: "Time Series Generation Under Data Scarcity: A Unified Generative Modeling Approach"
_NeurIPS.cc/2025/Conference — NeurIPS 2025 poster_

### Official Review · Reviewer_2Ztk · 2025-06-21

**Clarity:** 2
**Significance:** 3
**Originality:** 2
**Rating:** 4
**Confidence:** 4

**Summary:**

This paper introduces a time series generative model based on ImagenTime. To accommodate flexible number of channels, the authors introduced dynamic convolution and learnable data token as signals for labeling data domain. The model demonstrated particular advantage in data scarcity settings. At the same time, it claims to be the first pretrained time series generative model.

**Questions:**

line 216: each dataset is assigned a unique token that acts as an identifier of its source domain. How is this token assigned? Is it randomly chosen? Is it the domain name converted to word embeddings? There is no information on this.


There is an inconsistency in the metrics reported. some experiments/analysis use three metrics: discriminative, predictive, and contextFID while some experiments/analysis only reports discriminative and contextFID. why is that?

**Ethical Concerns:**

["NO or VERY MINOR ethics concerns only"]

**Final Justification:**

The authors provided additional exp on the challenging financial dataset evaluation where the corresponding dataset is removed from pretraining and still showed positive results. The authors also emphasized setting a standard for evaluation protocol.

**Limitations:**

The authors only mentioned one line at the end of section 6.1 saying the results on a few datasets like weather and ECG are challenging. But nothing is discussed about addressing the mentioned part. It could be due to a lack of datasets from the same domain in the pretraining set. Some analysis on similarity of pretraining and few shot dataset may help you uncover the reason. Additionally, some visualization on latent representation of learned dataset token during finetuning overlayed with dataset token specified during pretraining could be interesting.

**Quality:**

2

**Strengths And Weaknesses:**

Strength:
The modifications upon ImagenTime are intuitive.
The paper is easy to follow

Weakness:
The ablation study on dataset token did not show the advantage of dataset token. The effect of dataset token is not convincing and hard to interpret. Does that mean as long as dataset token is used in pretraining, it doesn't matter if it is used during finetuning? And are the benchmark dataset all from the domains seen during the pretraining? If not, do the dataset that come from an unseen domain show a different behavior when you look at the effect of dataset token? The experiment mentioned in line 345-246 is not shown in table 4. This is probably not the right way to report results.

I don't think Sine is a good dataset to include for evaluation, it might make more sense to be in the pretraining dataset. There is no distribution change for the sine dataset, so sine dataset does not pose any challenge in data-scarcity setting. The ETT datasets are also highly seasonal and probably not hard to learn in few shot settings. But stock dataset are hard but it was used in the pretraining dataset instead and there is no dataset from the financial domain in the evaluation.



It's probably not suitable to claim a systematic benchmark as a main contribution as there is only a few models compared. For example, TimeVAE which is more recent than TimeGAN is not compared. And data-scarcity setting might not be so overlooked as I remember there is such setting in the TimeVAE paper.

In figure 1A, it's only clear that your model performs better than baselines. But if your purpose it to show that the gap between #10 training samples and 100% training set for your model is smaller than baseline (drawn as x2.15), I don't think this goal is achieved. It's not obvious that the scaling behavior of your model is better than the orange and green line. You might reconsider what message should come cross.

---

> ### Author Rebuttal · Authors · 2025-07-30
>
> We thank Reviewer 2Ztk for the detailed and constructive review. We appreciate the recognition of our motivation and design choices, as well as the thoughtful questions and critiques regarding the dataset token, evaluation setup, benchmark scope, and clarity of presentation. These comments help identify areas where further explanation and refinement are needed. Below, we address each point and will revise the paper to improve clarity and better support our contributions.
>
> **W1**: *The ablation study on dataset token did not show the advantage of dataset token. The effect of dataset token is not convincing and hard to interpret. Does that mean as long as dataset token is used in pretraining, it doesn't matter if it is used during finetuning?*
>
> Thanks for highlighting this - we see that the motivation may not have been clearly conveyed.
>
> The dataset token addresses two main purposes. **Evaluate the pretrain** – By allowing the model to learn and differentiate between the multiple data sources, we can now reliably sample from a specific distribution during pretraining and evaluate the quality of pretraining. **Use an invested resource** – The pretraining phase is already complete, and the dataset token allows users to leverage this investment by sampling from the learned distribution.
>
> This is especially relevant in mixed distribution sampling. In fine-tuning, which involves a single distribution, the token holds little distributional value. However, we retain a task-specific token for architectural consistency. We’ll revise the manuscript to clarify this - thank you for your feedback.
>
> **W2**: *And are the benchmark dataset all from the domains seen during the pretraining?*
>
> Our evaluation includes both domains seen during pretraining and entirely new ones. For example, ETTm2 (fine-tuning) and ETTh1 (pretraining) both come from the electricity domain. In contrast, Mujoco (physics) and AirQuality (nature) are used only during fine-tuning, with no domain coverage in pretraining.
>
> **W3**: *If not, do the dataset that come from an unseen domain show a different behavior when you look at the effect of dataset token?*
>
> The dataset token has little effect on these datasets in the context of fine-tuning, which involves tuning the networks weights towards this distribution.
>
> **W4**: *The experiment mentioned in line 345-346 is not shown in table 4*
>
> The results in Lines 345–346 refer to the average performance across the pretraining datasets for the model without dataset token guidance, evaluated with 100% of the data and before fine-tuning. These are not shown in Table 4, which reports only fine-tuned downstream results.
>
> The full per-dataset results are provided in Table 18 (Appendix C.4). The reported values-0.252 (Disc.) and 11.07 (context-FID) without guidance, vs. 0.193 and 4.84 with guidance, are averages across the datasets in that table. We will revise the text to clarify this and include a direct reference to Table 18.
>
> **W5**: *I don’t think Sine is a good dataset to include for evaluation ... there is no distribution change for the sine dataset, so sine dataset does not pose any challenge in data-scarcity setting.*
>
>
> We included the synthetic sine dataset as a controlled benchmark to assess few-shot generation under low-noise, structured dynamics. While not reflective of real-world complexity, it serves as a sanity check for generalizing periodic behavior, as done in prior work (e.g., TimeGAN, ImagenTime).
>     Despite the lack of distribution shift, all models show a clear performance drop from 100% to limited data settings (Table 17), highlighting the challenge of few-shot generation even on simple signals.
>
> **W6**: *The ETT datasets are also highly seasonal and probably not hard to learn in few shot settings. But stock dataset are hard but it was used in the pretraining dataset instead and there is no dataset from the financial domain in the evaluation.*
>
> Thank you for raising this valid concern. The inclusion of the Stocks (and Energy) dataset in pretraining was intentional—we aimed to expose the model to data from variety of tasks, including time series generation, with Stocks (and Energy) serving as representative examples. To directly address your concern, we conducted a new experiment in which both the Stocks and Exchange datasets were removed from the pretraining corpus—effectively eliminating all financial data from pretraining. These datasets were then included to the fine-tune group, and the benchmark was re-run accordingly.
>
> The following tables show results for just Stocks and Exchanges dataset. Due to limited response space we comapred with second best model only. We observe that our model outperforms ImagenTime on all metrics on all setups on all datasets, even though no financial data was available in the pretrain phase.
>
> Table 4:
> Stocks:
> |%|metric|Our[24]|ImagenTime|
> |-|-|-|-|
> |5%|Disc|0.018|0.204|
> ||Pred|0.037|0.040|
> ||cFID|0.132|0.846|
> |10%|Disc|0.017|0.306|
> ||Pred|0.037|0.040|
> ||cFID|0.090|0.742|
> |15%|Disc|0.018|0.316|
> ||Pred|0.037|0.039|
> ||cFID|0.054|0.691|
> |10#|Disc|0.138|0.275|
> ||Pred|0.041|0.049|
> ||cFID|0.379|3.267|
> |25#|Disc|0.119|0.206|
> ||Pred|0.038|0.046|
> ||cFID|0.430|2.616|
> |50#|Disc|0.054|0.247|
> ||Pred|0.037|0.042|
> ||cFID|0.183|1.845|
>
> Exchange:
> |%|metric|Our[24]|ImagenTime|
> |-|-|-|-|
> |5%|Disc|0.046|0.470|
> ||Pred|0.522|0.564|
> ||cFID|0.137|3.552|
> |10%|Disc|0.036|0.47|
> ||Pred|0.520|0.564|
> ||cFID|0.119|3.366|
> |15%|Disc|0.044|0.473|
> ||Pred|0.522|0.559|
> ||cFID|0.124|3.285|
> |10#|Disc|0.352|0.496|
> ||Pred|0.606|0.611|
> ||cFID|2.574|8.13|
> |25#|Disc|0.279|0.476|
> ||Pred|0.604|0.599|
> ||cFID|2.222|5.901|
> |50#|Disc|0.178|0.491|
> ||Pred|0.564|0.578|
> ||cFID|0.833|4.639|
>
> **W7**: *It’s probably not suitable to claim a systematic benchmark as a main contribution as there is only a few models compared. For example, TimeVAE which is more recent than TimeGAN is not compared.*
> Thank you for the comment. Our goal is to support the standardization of evaluation protocols for few-shot time series generation. Given the wide range of generative models, we had to be selective due to computational and evaluation constraints. We chose methods based on two main criteria: (1) state-of-the-art performance and (2) architectural diversity. Accordingly, we included KoVAE (ICLR 2024, VAE-based), DiffTS and ImagenTime (ICLR/NeurIPS 2024, diffusion-based), and TimeGAN, a widely used GAN baseline. We would be happy to include additional models to further strengthen the benchmark’s scope and impact.
>
>
> Following your suggestion, we have now incorporated TimeVAE into our evaluation. The results are shown in the table below and will be included in the final version of the paper. We note our model out perform TimeVAE on all metrics and setups
>
> Table 5:
>
> |%|metric|Our[24]|TimeVAE|
> |-|-|-|-|
> |5%|Disc|0.11|0.239|
> ||Pred|0.458|0.467|
> ||cFID|0.674|1.747|
> |10%|Disc|0.083|0.232|
> ||Pred|0.452|0.466|
> ||cFID|0.578|1.692|
> |15%|Disc|0.066|0.215|
> ||Pred|0.451|0.465|
> ||cFID|1.086|1.303|
> |10#|Disc|0.259|0.312|
> ||Pred|0.489|0.51|
> ||cFID|3.8|2.838|
> |25#|Disc|0.19|0.26|
> ||Pred|0.467|0.473|
> ||cFID|1.582|1.874|
> |50#|Disc|0.149|0.239|
> ||Pred|0.46|0.475|
> ||cFID|0.987|1.75|
>
> **W8**: *In figure 1A, it's only clear that your model performs better than baselines. But if your purpose it to show that the gap between \#10 training samples and 100\% training set for your model is smaller than baseline (drawn as x2.15), I don't think this goal is achieved.*
>
> Thank you for the constructive feedback. The figure aimed to highlight the performance gap between the 100% and \#10 settings to support the benchmark’s validity, rather than emphasize our model’s superiority. We acknowledge that the 100%–\#10 gain for our model compared to other is unclear and that the \#50–100% jump may better illustrate this point. We will revise the figure and caption accordingly.
>
> **Q1**: *each dataset is assigned a unique token that acts as an identifier of its source domain. How is this token assigned?*
>
> Each dataset is assigned a unique identifier that serves as its token. The token is mapped to a learnable embedding, trained jointly with the model, and injected into the denoising network via AdaGN to condition generation on dataset identity.
>
> **Q2**: *There is an inconsistency in the metrics reported ...*
>
> Due to the NeurIPS format and 9-page limit, we were unable to include all metrics in Table 2 of the main paper and some metrics where moved to the appendix. For the full results, including Discriminative Score, Predictive Score, and context-FID, we refer to table 3 in reviewer zZSF respons (due to limited reponse space). We will include the full version of this table in the final paper.
>
> **Limitations 1**: *the results on a few datasets like weather and ECG are challenging. But nothing is discussed about addressing the mentioned part. It could be due to a lack of datasets from the same domain in the pretraining set.*
>
>
> Thank you for raising this important point—it has been a concern for us as well. We examined the link between underperformance and pretraining coverage using mutual information and FFT alignment. While Weather aligned well with the pretraining average and ECG200 did not, both showed poor results, revealing no clear pattern. Notably, ECG200 performed poorly despite ECG5000 (also ECG) being part of pretraining, suggesting domain similarity alone isn’t sufficient. Understanding these discrepancies remains an open and critical challenge, which we plan to investigate further and briefly discuss in the main paper.
>
> **Limitations 2**: *Additionally, some visualization on latent ... pretraining could be interesting*
>
> Visualizing the dataset tokens can indeed provide useful insights. We will include t-SNE and PCA plots in the paper, which show clear separation between tokens with noticeable margins. Some tokens—like those from Stocks and Exchange—reflect some domain similarity. We will include these results in the final version.

---

> > ### Comment · Reviewer_2Ztk · 2025-08-05
> > **Rebuttal Response**
> >
> > Thank you for the rebuttal response and additional experiments. You have mostly addressed my concern and I have raised my score.

---

### Official Review · Reviewer_WSzz · 2025-06-24

**Clarity:** 2
**Significance:** 2
**Originality:** 1
**Rating:** 2
**Confidence:** 5

**Summary:**

This paper is motivated by a key observation: even state-of-the-art generative models tend to suffer significant performance degradation in low-resource settings.

To address this challenge, the paper proposes a diffusion-based generative framework trained with a unified strategy, aiming to learn generalizable temporal representations that can bridge this performance gap. Specifically, it introduces two core components: (i) a dynamic convolution layer (DyConv) to handle channel mismatches across heterogeneous time series datasets during large-scale pretraining, and (ii) dataset label conditioning to guide the sampling process.

The paper underscores the importance of few-shot generative modeling in time series research and provides a step toward addressing this underexplored challenge.

**Questions:**

## Questions
Baseline selection: The paper compares against ImagenTime, DiffTS, KoVAE, and TimeGAN. While these baselines span a broad range of generative paradigms, some—like KoVAE and TimeGAN—are relatively classical. It's unclear whether these baselines represent current state-of-the-art performance.

Clarity issues remain unresolved, especially those mentioned earlier regarding model explanation.

Dataset label conditioning ablation: The paper states that models trained without dataset label guidance experience significant performance drops if not fine-tuned. However, it's unclear where this result is reported. If a corresponding table exists, please point to it explicitly.

Metric averaging ambiguity: In Section 6.1 on few-shot benchmarks, Table 1 reports the average values of Discriminator Score (Disc.), Predictor Score (Pred.), and context-based FID (c-FID). However, it is unclear whether these averages refer to the mean across different domains or the mean of multiple runs for each metric. This ambiguity could lead to misinterpretation.

Terminology inconsistencies: For instance, describing the model’s ability to process variable-length time series using dynamic masks as "variable-length time series generation" is somewhat imprecise. Similarly, describing the experiment in Section 6.2—on pretraining with different sequence lengths—as “cross-resolution transfer” may be inaccurate.

**Ethical Concerns:**

["NO or VERY MINOR ethics concerns only"]

**Final Justification:**

Thank you very much to the authors for the clarifications provided on several issues. However, I still find some of the key conclusions of the paper unconvincing or not particularly compelling.

1. The framework builds upon ImagenTime (2024) and improves transferability through DyConv and label conditioning. Nevertheless, the overall level of innovation remains limited.

2. I appreciate the authors' supplementary experiments and clarifications regarding the baselines. The inclusion of TimeVAE helps broaden the benchmark coverage to some extent. However, this addition does not fully address my previous concern: although the selected baselines span different paradigms, some are outdated and no longer representative of the current state-of-the-art. In particular, TimeVAE, introduced at ICLR 2022, is itself relatively dated. Thus, It is very hard to validate the effectiveness of your proposal.

3. Furthermore, since the proposed method is based on a diffusion model framework, incorporating more recent diffusion-based time series generation models as baselines would make the evaluation more convincing. Including such models would allow for a more accurate comparison of performance and a clearer understanding of the contribution to the rapidly evolving field of time series generation.

4. The evaluation process in TSGBench (VLDB 2024) has become fairly standardized, with 29 citations, which is relatively high in the time series generation community and reflects its recognition for benchmarking purposes. The current evaluation in this paper lacks several key metrics (e.g., MDD, ACD, SD, KD, ED, DTW), making it difficult to comprehensively assess the model’s generalization and performance. We encourage the authors to incorporate these metrics in future revisions to enhance the thoroughness of the evaluation.

4. Regarding the previous concern about the ambiguity of average metrics, I thank the authors for their clear and detailed clarification. Although the appendix includes the complete results for each dataset and subset size, reporting the average performance across all domains as the overall result in the main text still lacks justification.

5. I appreciate the authors’ thorough responses to all concerns. The clarifications in the rebuttal are likely to improve the manuscript’s readability. However, I still feel that the overall contribution to NeurIPS is somewhat limited. The paper essentially applies the ImagenTime model from NeurIPS 2024 to the task of general time series generation with pretraining, and introduces DyConv and label conditioning to enable transferability. In my view, this mainly constitutes a combination of existing methods.

Given the issues outlined above, I believe the paper still requires substantial revisions. Therefore, I will maintain my original score.

**Limitations:**

yes

**Quality:**

2

**Strengths And Weaknesses:**

## Quality
The paper presents a series of designs to adapt the ImagenTime model for large-scale unified generative pretraining. In particular, DyConv is proposed to address the issue of misaligned channels across multi-domain heterogeneous time series data, and a masking mechanism is introduced to handle variable-length time series converted into images. These design choices are generally reasonable.

The ablation study for DyConv is fair. The authors remove the DyConv module and replace it with static channel padding as a baseline. To ensure fair comparison, a masking mechanism is incorporated into the loss function so that the denoising network is only penalized on valid (unpadded) dimensions. This improves the credibility of the ablation results.

## Clarity
The explanation of the loss function (Equation 1) is somewhat unclear, particularly regarding how the weight function $\lambda(\sigma)$ is defined.

The model introduces a dynamic binary mask applied to the padded input sequence, which is adjusted at runtime to distinguish between valid and padded time steps. While this mask design is reasonable for handling variable-length time series converted into images, the paper fails to explain how this dynamic adjustment is implemented.

The section on dataset label conditioning states that the label is mapped to a learnable embedding and injected into the denoising network via Adaptive Group Normalization (AdaGN) [17] during training and generation. However, the implementation details are vague—this part would benefit from a formula or schematic illustration.

## Significance
The paper clearly highlights an important issue: although generative modeling is a promising approach to mitigating data scarcity, existing models are typically designed and evaluated under the assumption of abundant training data. This makes the paper relevant to the time series generation community.

## Originality
The proposed framework builds upon ImagenTime, a 2024 time series generative model that transforms time series into image-shaped inputs to leverage diffusion models from the vision domain. While the paper justifies using ImagenTime as the backbone and introduces DyConv and dataset label conditioning for transferability, the overall novelty remains somewhat limited.

---

> ### Author Rebuttal · Authors · 2025-07-30
>
> We thank Reviewer WSzz for the thoughtful and detailed review. We appreciate the recognition of our motivation, as well as the design contributions such as DyConv and dataset label conditioning. We found the questions and suggestions - particularly those regarding implementation clarity, evaluation protocol, and terminology, very helpful. These insights have allowed us to refine our explanations and improve the presentation. Below, we address each point in detail and will revise the paper to incorporate the reviewer’s feedback.
>
> **W1**: *The explanation of the loss function (Equation 1) is somewhat unclear*
>
> We are happy to provide more detail. Generally, our approach to defining the loss function for our diffusion process follows the methodologies presented in EDM [1] and ImagenTime [2].
> $$
> \mathcal{L} = \mathbb{E} _ {\sigma,x _ \text{img}^0,\varepsilon} [ \lambda(\sigma) \left\| {\Gamma_\theta (x_\text{img}^\sigma;\sigma)-x _ \text{img}^0} \right\| _ 2^2 ] \ ,
> $$
> where $\lambda(\sigma)$ is a weighting function and $\Gamma_\theta$ is defined as follows:
> $$
> \Gamma _ \theta (x _ \text{img}^\sigma; \sigma) = c_{\text{skip}}(\sigma)x_\text{img}^\sigma + c_\text{out}(\sigma) N_{\theta}(c_\text{in}(\sigma)x_\text{img}^\sigma;c_{\text{noise}}(\sigma);y) \ .
> $$
> In this equation, $N_{\theta}$ represents the denoiser neural network. The terms $c_\text{skip}(\sigma)$, $c_\text{in}(\sigma)$, $c_\text{out}(\sigma)$, and $c_\text{noise}(\sigma)$ are all functions of $\sigma$.
>
> - $c_\text{skip}(\sigma) = \frac{\sigma_\text{data}^2}{\sigma^2 + \sigma_\text{data}^2}$
> - $c_\text{out}(\sigma) = \frac{\sigma \cdot \sigma_\text{data}^2}{\sqrt{\sigma^2 + \sigma_\text{data}^2}}$
> - $c_\text{in}(\sigma) = \frac{1}{\sqrt{\sigma^2 + \sigma_\text{data}^2}}$
> - $c_\text{noise}(\sigma) = \frac{1}{4} \ln(\sigma)$
>
> We use $\sigma_\text{data} = 0.5$ as a fixed parameter, and $\sigma(t)$ controls the noise level over time. These terms help stabilize the loss, which can vary with $\sigma$.
>
> Substituting Eq.\~2 into Eq.\~1 yields a per-sample loss weight. To balance this, EDM sets $\lambda(\sigma) = 1/c_\text{out}(\sigma)^2$, also ensuring uniform initial loss across the $\sigma$ range.
>
> Though our focus wasn’t on loss design, we agree this clarification is helpful and will include it in the revised paper.
>
> **W2**: *The model introduces a dynamic binary mask applied to the padded input sequence, ... the paper fails to explain how this dynamic adjustment is implemented*
>
> We convert time series into images using delay embedding and zero-padding. To identify valid (non-padded) regions, we apply the same transformation to an all-ones tensor, creating a binary mask.
>
> During training, noise is added to the full image, but loss is computed only over valid pixels.
>
> “Dynamic adjustment” refers to how the valid region (and thus the mask) changes with input length, though image size remains fixed. Each dataset has a consistent length, but lengths differ across datasets. Section 6.2 shows how our model handles length mismatches during pretraining and fine-tuning. We will clarify this in the text and release code.
>
> **W3**: *The section on dataset label conditioning states that the label is mapped to a learnable embedding and injected into the denoising network via Adaptive Group Normalization. However, the implementation details are vague*
>
> We thank the reviewer for pointing this out. The Adaptive Group Normalization (AdaGN) implementation is as follows: $\text{AdaGN}(h,y) = y_s~\text{GroupNorm}(h) + y_b$ where $h$ is the intermediate activations of the residual block following the first convolution, and $y=[y_s,y_b]$ is obatined from a linear projection  of the timestep and the data token embedding.
>
> In our UNet architecture, AdaGN is applied within each block, allowing the model to condition its activations on both the timestep and dataset identity. For clarity, we include a simplified UNet block code snippet below highlighting this integration. We will expand on this explanation—including the formula—in the revised version and will release the code publicly.
>
> ``` python
> class UNetBlock(torch.nn.Module):
>     ...
>     def forward(self, x, emb):
>         orig = x
>         x = self.conv0(silu(self.norm0(x)))
>
>         # Project the embedding (timestep + data token)
>         params = self.affine(emb)
>         if self.adaptive_scale:
>             # Split parameters into scale and shift
>             scale, shift = params.chunk(chunks=2, dim=1)
>             # Apply AdaGN: scale * GroupNorm(x) + shift
>             x = silu(torch.addcmul(shift, self.norm1(x), scale + 1))
>         else:
>             # Fallback for non-adaptive scaling
>             x = silu(self.norm1(x.add_(params)))
>     ...
> ```
> **W4**: *While the paper justifies using ImagenTime as the backbone and introduces DyConv and dataset label conditioning for transferability, the overall novelty remains
> somewhat limited*
>
> We'd like to clarify and emphasize the innovative aspects of our work. Our core contribution lies in establishing the first robust baseline for time series generation in data-scarce environments.
>
> While we build on ImagenTime’s idea of converting time series into image-like inputs for diffusion models, our key innovation—shown in Figure 2—is a novel design not previously explored in time series generation. By bridging research on unified training with this domain, our approach offers a strong, versatile framework that outperforms existing methods, especially in low-data settings.
>
> We acknowledge that components like diffusion models and time series-to-image transforms aren't new, nor is the use of unified training. However, applying this paradigm to time series generation—with varying channel sizes and diverse training datasets—introduces unique challenges that our method addresses effectively. Interestingly, our unified training paradigm for time series enables analyses that are not possible in other settings, adding another novel dimension to our contribution.
>
> **Q1**: *Baseline selection: The paper compares ... While these baselines span a broad range of generative paradigms, some-like KoVAE and TimeGAN-are relatively classical...*
>
> Our chosen baselines represent a robust and state-of-the-art set of generative models for time series, drawing primarily from recent, top-tier conferences in the field:
>
> - **ImagenTime** published at NeurIPS 2024.
> - **DiffusionTS (DiffTS)** published at ICLR 2024.
> - **KoVAE**, published at ICLR 2024.
> - **TimeGAN**, while more classical, remains a widely-used benchmark for time series generation. We included TimeGAN to span diverse generative paradigms, encompassing GAN-, VAE-, and diffusion-based models.
>
> Additionally, we have incorporated **TimeVAE** into our evaluation for completeness. Due to limited reponse space and alignment with other reviewers, please refer to table 5 in reviewer 2Ztk response .We will include this updated table in the final version of the paper. We note our model outperform TimeVAE on all metrics and setups. We appreciate the feedback and the opportunity to clarify this point, as well as to strengthen our scarce data comparisons and benchmarking. We would be happy to address any further suggestions.
>
> **Q2**: *Dataset label conditioning ablation: The paper states that models trained without dataset label guidance experience significant performance drops if not fine-tuned. However, it’s unclear where this result is reported*
>
> This is a subtle but important point that we will clarify in the main text.
>
> We find that removing the dataset token during pre-training leads to a noticeable drop in evaluation metrics on the pre-training datasets. Specifically, without the token, the model achieves an average Discriminative Score of 0.252 and context-FID of 11.07. In contrast, with the token, scores improve to 0.193 and 4.84, respectively. Full results, including per-dataset comparisons, are shown in Table 18 (Appendix C.4). This demonstrates the token's key role in guiding sampling across heterogeneous datasets during pre-training.
>
> However, as seen in Table 4, the token is not needed during fine-tuning. Once the model is adapted to a specific dataset, the token’s original purpose fades. Still, it remains crucial when generating from mixed or unseen data distributions.
>
> We will revise the paper to make this distinction clear and avoid confusion.
>
> **Q3**: *Metric averaging ambiguity: In Section 6.1 on few-shot benchmarks ... However, it is unclear whether these averages refer to the mean across different domains or the mean of multiple runs for each metric*
>
> We agree that the averaging procedure should have been clarified more explicitly. The reported scores in Table 1 are averaged across all evaluation datasets for each subset size. For example, in the 100\% row, each model's score reflects the average performance across all domains under the full-data setting. Similarly, in the 10\% row, the reported scores are the averages across all domains under the 10\%-data setting. For a complete picture, we've included the full, per-dataset, per-subset size results in the appendix.
>
> We will incorporate this clarification into the paper.
>
> **Q4**: *Terminology inconsistencies: For instance, describing the model’s ability to process variable-length time series using dynamic masks as "variable-length time series generation" is somewhat imprecise. Similarly, describing the experiment in Section 6.2—on pretraining with different sequence lengths—as “cross-resolution transfer” may be inaccurate.*
>
> We acknowledge that both terms may be unclear and imprecise. We will revise them to "cross-temporal resolution transfer learning," which more accurately describes how our model transfers knowledge across different temporal resolutions - training on one and fine-tuning on another, as demonstrated in Section 6.2. This change will be reflected in the revised paper to improve clarity. Finally, we will use time series generation of variable-length inputs instead.

---

> > ### Comment · Reviewer_WSzz · 2025-08-05
> >
> > Thank you very much to the authors for the clarifications provided on several issues. However, I still find some of the key conclusions of the paper unconvincing or not particularly compelling.
> >
> > 1. The framework builds upon ImagenTime (2024) and improves transferability through DyConv and label conditioning. Nevertheless, the overall level of innovation remains limited.
> >
> > 2. I appreciate the authors' supplementary experiments and clarifications regarding the baselines. The inclusion of TimeVAE helps broaden the benchmark coverage to some extent. However, this addition does not fully address my previous concern: although the selected baselines span different paradigms, some are outdated and no longer representative of the current state-of-the-art. In particular, TimeVAE, introduced at ICLR 2022, is itself relatively dated. Thus, It is very hard to validate the effectiveness of your proposal.
> >
> > 3. Furthermore, since the proposed method is based on a diffusion model framework, incorporating more recent diffusion-based time series generation models as baselines would make the evaluation more convincing. Including such models would allow for a more accurate comparison of performance and a clearer understanding of the contribution to the rapidly evolving field of time series generation.
> >
> > 4. The evaluation process in TSGBench (VLDB 2024) has become fairly standardized, with 29 citations, which is relatively high in the time series generation community and reflects its recognition for benchmarking purposes. The current evaluation in this paper lacks several key metrics (e.g., MDD, ACD, SD, KD, ED, DTW), making it difficult to comprehensively assess the model’s generalization and performance. We encourage the authors to incorporate these metrics in future revisions to enhance the thoroughness of the evaluation.
> >
> > 4. Regarding the previous concern about the ambiguity of average metrics, I thank the authors for their clear and detailed clarification. Although the appendix includes the complete results for each dataset and subset size, reporting the average performance across all domains as the overall result in the main text still lacks justification.
> >
> > 5. I appreciate the authors’ thorough responses to all concerns. The clarifications in the rebuttal are likely to improve the manuscript’s readability. However, I still feel that the overall contribution to NeurIPS is somewhat limited. The paper essentially applies the ImagenTime model from NeurIPS 2024 to the task of general time series generation with pretraining, and introduces DyConv and label conditioning to enable transferability. In my view, this mainly constitutes a combination of existing methods.
> >
> > Given the issues outlined above, I believe the paper still requires substantial revisions. Therefore, I will maintain my original score.

---

> > > ### Author Response · Authors · 2025-08-08
> > > **Discussion**
> > >
> > > ***(P1 + P6)***  We respectfully disagree with the characterization of our work as primarily a combination of existing methods. Our core contribution is not just an architectural refinement, but the identification and solution of a critical, unaddressed problem in time series generation: the significant performance degradation of state-of-the-art models in data-scarce environments.
> > >
> > > To our knowledge, our work is the first to:
> > >
> > > - Conduct a large-scale empirical study that quantifies this performance gap across leading generative models.
> > >
> > > - Demonstrate that existing SOTA models often fail to generalize when training data is limited.
> > >
> > > - Propose a simple yet highly effective pre-training framework, incorporating Dynamic Convolutions (DyConv) and dataset tokens, that specifically bridges this performance gap and enables robust transferability.
> > >
> > > We believe that identifying this critical problem and providing an effective solution represents a substantial and timely contribution to the field.
> > >
> > > ***(P2 + P3)*** We aimed to compare against a representative and recent set of SOTA methods. Our comparison includes ImagenTime (NeurIPS 2024), Diff-TS (ICLR 2024) and KoVAE (ICLR 2024), which are very recent and two of them are prominent diffusion-based models.
> > >
> > > The inclusion of TimeVAE was in direct response to a suggestion from another reviewer to further strengthen the benchmark’s scope and impact. If the reviewer has specific recent models in mind they believe are critical for comparison, we would be grateful for the suggestion.
> > >
> > >
> > > ***(P4)*** We thank the reviewer for highlighting TSGBench and its growing role in the community. We appreciate the suggestion to include additional metrics such as MDD, ACD, SD, KD, ED, and DTW, and we agree that incorporating a broader set of evaluation criteria can help paint a more comprehensive picture of model performance.
> > >
> > > However, we would like to respectfully note that this concern was not raised in the initial review, and thus we did not have the opportunity to address it earlier in the discussion. At this stage, when the discussion period is concluding, introducing entirely new evaluation expectations makes it difficult for us to respond appropriately within the process timeline. That said, we view this as valuable feedback and will consider integrating TSGBench metrics in future iterations of our work.
> > >
> > > ***(P5)*** We are open to constructive suggestions. However, we respectfully disagree with the suggestion that reporting average performance across all domains in the main text “lacks justification.” This practice is widely adopted in the time series and broader machine learning communities, where averaged metrics serve as a concise and interpretable summary of overall model performance. Full disaggregated results are provided in the appendix to ensure transparency and allow for in-depth analysis.
> > >
> > > This is a direct consequence of the strict NeurIPS page limits, which make it infeasible to include exhaustive per-dataset results in the main paper. That said, if the reviewer has seen effective strategies in other publications for presenting such granular data within these constraints, we would be very grateful for their guidance.

---

### Official Review · Reviewer_zZSF · 2025-06-24

**Clarity:** 2
**Significance:** 3
**Originality:** 4
**Rating:** 6
**Confidence:** 5

**Summary:**

Proposes a new method for training unconditional time-series generation models by pretraining on heterogeneous time series data and finetuning on data from the target domain. For evaluation, it introduces a benchmark which evaluates different models under varying degrees of data scarcity.

**Questions:**

1. What dataset is being used to compared with in Table 1? Is the synthetic data trained on 5% of the data compared to 100% of the data, or is it compared to the 5% subset?
2. What do you think about the originality of the synthetic data itself? Is it copies of the training data, or something more? It could be worthwhile to evaluate this empirically (e.g. using [1]), especially if the answer to the previous questions is that the synthetic data trained on 5% is compared to the 5% subset.
3. Why is predictive score not presented for the model size study in Table 2?
4. Could you elaborate a bit on the similarities and differences between DynConv and DyLinear?
5. How are the datasets divided into several sequences? E.g. the stocks dataset consists of a single sequence.

**Ethical Concerns:**

["NO or VERY MINOR ethics concerns only"]

**Final Justification:**

The authors have responded to my concerns about evaluation (Q1, Q2, Q3, and Q5) and provided additional details on their methodology (Q4). This paper addresses the critical challenge of data scarcity, which is especially relevant to the field of time-series. If accepted, this can open up a new direction for future time-series generative model research with even more efficient use of the available data.

**Limitations:**

yes

**Paper Formatting Concerns:**

No concerns.

**Quality:**

3

**Strengths And Weaknesses:**

**Strengths:**

- The problem is clearly defined and thoroughly motivated
- The model is compared to a wide range of pre-existing unconditional time-series generation models under varying degrees of data scarcity
- The pretraining is also demonstrated to allow the model to better generalize across different sequence lengths
- The ablation study evaluates the impact of two critical components of the approach
- Makes valuable improvements to the ImagenTime architecture to accomodate multi-domain pretraining:
	- Dataset token
	- Dynamic convolution to handle different datasets having different number of dimensions per time step
- Provides new insights into how to make the most out of the limited data available in unconditional time-series generation. Although pretraining techniques are common in other domains,they have not been thoroughly explored within the context of unconditional time-series generation. Ultimately, this might open up for generation of new domains in time-series which were previously inaccessible due to limited data.

**Weaknesses:**

- The paper focuses on relatively short sequence lengths, using sequence length 32 for the main results in Table 2, which limits real world use. This can be compared to the ImagenTime paper, which generates for up to 17k in their longest experiments.
- Technical details that should be clarified:
	- What dataset is being compared to when using c-FID, discrimination score and predictive score in for the various scarcity levels? Is it the scarce variant? Or the complete dataset? Depending on the choice here, different conclusions might be drawn. If the scarce datasets are compared to the synthetic data, there is a risk that the model has learned to memorize the training data and produce copies. This would lead to very good scores, but the samples would just be copies of the original training data. There are evaluation measures that aim to measure the degree of originality of synthetic data, see e.g. [1].
	- The predictive score is not presented for model size ablation in Table 2

[1] Alaa, A., Van Breugel, B., Saveliev, E.S., and van der Schaar, M.. (2022). How Faithful is your Synthetic Data? Sample-level Metrics for Evaluating and Auditing Generative Models. Proceedings of the 39th International Conference on Machine Learning, in Proceedings of Machine Learning Research 162:290-306 Available from https://proceedings.mlr.press/v162/alaa22a.html.

---

> ### Author Rebuttal · Authors · 2025-07-30
>
> We thank Reviewer zZSF for the thoughtful and detailed review. We appreciate the recognition of our contributions, including the multi-domain pretraining setup, architectural improvements to ImagenTime, and the proposed benchmark for data-scarce generation. We also value the reviewer’s insightful questions regarding evaluation protocol, model behavior, and design decisions, which helped us clarify and strengthen the paper. Below, we address each point and would be glad to incorporate this feedback into a final revision.
>
> **W1**:  *The paper focuses on relatively short sequence lengths, using sequence length 32 for the main results in Table 2, which limits real world use. This can be compared to the ImagenTime paper, which generates for up to 17k in their longest experiments*
>
> This is a good suggestion and we agree that extending time-series generation to long sequences is a crucial challenge for many real-world applications. Our primary goal was to investigate performance across diverse generative paradigms (GAN, VAE, Diffusion) in data-scarce settings. To ensure a fair and broad comparison, we chose a sequence length compatible with all baseline architectures, as models like TimeGAN, KoVAE, and DiffusionTS were not originally designed for long sequences and would require significant modifications. Focusing only on long sequences would have narrowed our analysis to a single method (i.e., ImagenTime), preventing the cross-paradigm study we intended. We agree that exploring our unified framework for long-sequence generation is a key direction for future work and we will add a short discussion about it in the new revision.
>
> **W2.1**: *What dataset is being compared to when using c-FID, discrimination score and predictive score in for the various scarcity levels? Is it the scarce variant?*
>
>  *What dataset is being used to compared with in Table 1? Is the synthetic data trained on 5% of the data compared to 100% of the data, or is it compared to the 5% subset?*
>
>  *What do you think about the originality of the synthetic data itself? is it copies of the training data, or something more*
>
> The training dataset is a subset of the original data - depending on the setup (i.e., #10,..,5%,..), while the test dataset always comprises the full original dataset (i.e., 100\%). We will revise the paper to make this distinction clearer. We appreciate your concern regarding potential memorization; however, given the model’s limited access to the data - especially relative to the full dataset, we believe the risk of memorization is minimal. To further support this point, we will include t-SNE and PCA visualizations of some of the fine-tuned representations in the appendix. In summary, the qualitative t-SNE/PCA results show that the generated distributions closely overlap with the input distributions while still exhibiting meaningful variability. Additionally, we plan to discuss [1] in the final manuscript to highlight the importance of assessing the originality of synthetic data.
>
> **W2.2**: *The predictive score is not presented for model size ablation in Table 2*
>
> Due to the NeurIPS format and 9-page limit, we were unable to include all metrics in Table 2 of the main paper, and reported it in the appendix. The full results, including Discriminative Score, Predictive Score, and context-FID, are shown in the complete table below. We will include the full version of this table in the final paper.
>
> Table 3:
>
> |Metric| |Base|Medium|Large|XL|XL→S Det.|
> |-|-|-|-|-|-|-|
> |Disc|FT|0.15| 0.16| 0.13| 0.13|-14.02%|
> ||w/o PT| 0.35| 0.35 |0.26|0.27|-23.29%|
> | cFID| FT |4.85| 5.03|4.56|4.51|-7.37%|
> ||w/o PT |15.87 |15.36|8.12|8.15|-48.62% |
> |Pred| FT | 0.4993|0.4974|0.4959|0.4965|**-0.57%**|
> ||w/o PT|0.5043|0.5036|0.5007|0.5023|**-0.40%**|
>
> **Q4**: *Could you elaborate a bit on the similarities and differences between DynConv and DyLinear?*
>
> We believe the motivation behind DyLinear was to enable a single model to support time series of multiple lengths. The main challenge arises from the fact that model instantiation occurs during pretraining, at which point we have no prior knowledge of the fine-tuning dataset, specifically, its shape. In other words, we must initialize and train model weights without knowing the final input dimensions.
>
> DyLinear addresses this cleverly by interpolating between canonical weights and the target shape required for the linear operation. Once interpolated, the layer performs a standard linear transformation using the adapted weights and the input.
>
> We extend this idea to the convolutional setting: we define a single canonical convolution kernel and apply interpolation to reshape it according to the target dimensions. The resulting kernel is then used in the standard convolution operation.
>
> **Q5**: *How are the datasets divided into several sequences?*
>
> A major advantage of time series generation is that data from diverse tasks can often be repurposed for generative modeling. However, instead of applying a one-size-fits-all rule, we adopted dataset-specific preprocessing strategies (e.g., as done in TimeGAN and UCR/UEA benchmarks) to remain faithful to prior work and ensure reproducibility.
>
> For all datasets, we used the original preprocessing protocols defined in their respective benchmarks. For instance, the Stocks dataset from TimeGAN was segmented using the same sliding window method as in the original implementation.

---

> > ### Comment · Reviewer_zZSF · 2025-08-06
> >
> > I would like to thank the authors for taking the time to read and respond to my questions. The concerns I raised in my review have all been sufficiently addressed.

---

### Official Review · Reviewer_g6T2 · 2025-07-03

**Clarity:** 3
**Significance:** 4
**Originality:** 3
**Rating:** 5
**Confidence:** 4

**Summary:**

This paper proposes a unified generative modeling framework for time series generation that is pre-trained across a diverse set of datasets spanning multiple domains (e.g., finance, climate, energy, biomedical) and fine-tuned for specific target domains under low-data conditions. It uses a vision-based diffusion model that transforms time series into image representations, incorporates dynamic convolution (DyConv) layers to flexibly handle varying input/output channel sizes, and uses dataset-specific tokens to guide generation. The model achieves strong few-shot performance, supports variable sequence lengths, and shows robustness across model sizes and domains.

**Questions:**

1. Why was delay embedding chosen over sequence-native encodings for diffusion modeling?

2. What is the impact of the delay embedding parameters (e.g., window size) on generation quality?

3. How robust is the dataset token mechanism when domain boundaries are ambiguous or overlapping?

4. Does the image transformation degrade interpretability of the generated sequences?

5. Could lightweight alternatives to DyConv (e.g., grouped or depthwise convolution) provide similar flexibility with lower cost?

6. How sensitive is performance to the number and diversity of datasets used in pretraining?

7. Are there cases where pretraining leads to negative transfer in fine-tuning?

**Ethical Concerns:**

["NO or VERY MINOR ethics concerns only"]

**Limitations:**

The authors acknowledge some dataset-specific limitations in performance (e.g., on Weather and ECG200), but a deeper analysis into why the model underperforms on these datasets is missing. Understanding whether the challenges stem from noise, irregular sampling, or lack of temporal structure could provide more actionable insights. Additionally, while the model spans high-stakes domains like finance and biomedical data, the discussion on potential societal risks—such as misuse of synthetic data, overreliance on generated samples, or downstream model degradation due to hallucinations—is minimal. The paper would benefit from explicitly addressing these concerns and proposing guardrails for responsible use.

**Quality:**

3

**Strengths And Weaknesses:**

Strengths
The unified framework tackles a major limitation of prior generative models by enabling generalization across domains and strong performance in few-shot settings, outperforming state-of-the-art models like ImagenTime and DiffusionTS by large margins. Its use of delay-embedding image transformations leverages advances in vision diffusion models, while DyConv and dataset tokens offer architectural flexibility, efficiency, and improved fidelity across multivariate time series with varying lengths and channels. The comprehensive evaluation across datasets, sequence lengths, and model scales strengthens the validity of its claims.

Weaknesses
Despite the gains, the approach relies on image-based representations, which may introduce unnecessary inductive biases or miss native temporal dependencies compared to sequence-native architectures. The use of pretraining and fine-tuning still requires nontrivial computational resources and tuning. Additionally, while dataset tokens improve specificity, their utility diminishes without fine-tuning, and performance on some domains (e.g., Weather, ECG200) still lags in limited data settings, suggesting room for better domain adaptation or representation alignment.

---

> ### Author Rebuttal · Authors · 2025-07-30
>
> We thank Reviewer g6T2 for their thoughtful and constructive review. We appreciate the recognition of our framework’s strengths, including its strong few-shot performance, cross-domain generalization, and architectural choices such as DyConv and dataset tokens. We also value the insightful questions and suggestions regarding interpretability, robustness, and limitations. Below, we address each point in detail and would be happy to incorporate this feedback in a final revision.
>
> **W1 + Q1:**  *the approach relies on image-based representations, which may introduce unnecessary inductive biases or miss native temporal dependencies compared to sequence-native architectures. Why was delay embedding chosen over sequence-native encodings for diffusion modeling?*
>
> Our method extends the ImagenTime framework, which converts time series data into image representations to achieve state-of-the-art results.  While representing time series as images may introduce inductive biases that could restrict the model’s ability to fully capture temporal dynamics, our experiments show that our model consistently delivers the most reliable and highest-quality generations. To further validate this, we conducted additional evaluations following the setup of Experiment 6.1, using sequence-native models like DiffusionTS and KoVAE as pretrained baselines. As illustrated in the table below, both baselines consistently underperform compared to our approach.
>
> Table 1:
>
> |percentage|metric|Our[24]|DiffTS|KoVAE|
> |-|-|-|-|-|
> |**5%**|Disc|**0.11**|_0.244_|0.331|
> ||Pred|**0.458**|_0.471_|0.5|
> ||cFID|**0.674**|_2.668_|3.719|
> |**10%**|Disc|**0.083**|_0.223_|0.342|
> ||Pred|**0.452**|_0.464_|0.491|
> ||cFID|**0.578**|_2.432_|3.874|
> |**15%**|Disc|**0.066**|_0.223_|0.338|
> ||Pred|**0.451**|_0.466_|0.486|
> ||cFID|**1.086**|_2.475_|3.549|
> |**10#**|Disc|**0.259**|_0.347_|0.375|
> ||Pred|**0.489**|_0.516_|0.517|
> ||cFID|**3.8**|_4.702_|5.522|
> |**25#**|Disc|**0.19**|_0.31_|0.352|
> ||Pred|**0.467**|_0.492_|0.499|
> ||cFID|**1.582**|_3.817_|3.844|
> |**50#**|Disc|**0.149**|_0.287_|0.354|
> ||Pred|**0.46**|_0.48_|0.491|
> ||cFID|**0.987**|_3.116_|4.001|
>
> **W2**:  *The use of pretraining and fine-tuning still requires nontrivial computational resources and tuning.*
>
> Our pretraining takes 4 training hours of 2 NVIDIA RTX-4090. This may be converted into a single RTX-4090 with $\sim 8$ hours of training. Assuming the available pretrain model checkpoint, the finetune process itself takes on average only $\sim 15$ minutes on a single GPU.
>
> **Q2**:  *What is the impact of the delay embedding parameters (e.g., window size) on generation quality?*
>
> The original ImagenTime paper offers an in-depth analysis of delay embedding, outlining its advantages and limitations. Importantly, it includes an ablation study showing that delay embedding is robust across different window sizes and generally outperforms alternatives such as STFT.
>
> **Q3**:  *How robust is the dataset token mechanism when domain boundaries are ambiguous or overlapping?*
>
> To evaluate the robustness of the dataset token in settings with closely related distributions, we pre-trained two versions of the model - one with the dataset token and one without, using only the ETT family (ETTh1/h2, ETTm1/m2).
> We then evaluated the pre-trained models on the same ETT datasets to assess how well the token helps disambiguate subtle inter-dataset variations.
> Results in the table below show a significate improvement when sampling with dataset token opposed to without even when domain boundaries are ambiguous / overlapping.
>
> Table 2:
>
> |Dataset|Metric|w/ dataset token|w/o dataset token|
> |-|-|-|-|
> |ETTh1|Disc|0.033|0.344|
> ||Pred|0.646|0.674|
> ||cFID|0.108|5.687|
> |ETTh2|Disc|0.027|0.263|
> ||Pred|0.681|0.707|
> ||cFID|0.086|3.360|
> |ETTm1|Disc|0.007|0.232|
> ||Pred|0.675|0.690|
> ||cFID|0.0210|2.920|
> |ETTm2|Disc|0.007|0.236|
> ||Pred|0.694|0.718|
> ||cFID|0.0249|1.248|
>
> **Q4**:  *Does the image transformation degrade interpretability of the generated sequences?*
>
> No it does not. In our work, we use an invertible transofmation to convert time series into image-shaped inputs. This means that the process of converting the time series to an image representation and then back to the original signal representation is lossless. This ensurs full interpretability of the generated sequences, inverting the image back to time-series without loosing information.
>
> **Q5**:  *Could lightweight alternatives to DyConv (e.g., grouped or depthwise convolution) provide similar flexibility with lower cost?*
>
> Grouped or depthwise convolutions indeed reduce compute, but they cannot dynamically adapt a single shared kernel across arbitrary input/output channel configurations. Depthwise convolution processes channels independently (groups = input channels), and even combined with pointwise operations it still relies on fixed Cin/Cout sizes. In contrast, DyConv uses bicubic-interpolated resizing of a canonical weight tensor to match any channel dimensions at runtime, which enables seamless generalization across diverse multivariate datasets
>
> **Q6**:  *How sensitive is performance to the number and diversity of datasets used in pretraining?*
>
> This is an interesting question that we considered during our research, but it remains open and represents an important direction for future work. As it falls outside the scope of our current study, we do not yet have a definitive answer. We thank the reviewer for raising this point and will incorporate a brief discussion in the main paper to highlight it as a valuable open question for future investigation.
>
> **Q7**:  *Are there cases where pretraining leads to negative transfer in fine-tuning?*
>
> Our extensive empirical investigation shows that, in the majority of cases, it does not. In Figure 3A (Experiment 6.2) and Table 2 (Experiment 6.3), we show that the pre-trained model outperforms the non-pretrained model on average across datasets. To complement this, we also evaluated individual datasets. Across all cases, pretraining consistently improves both the Discriminative Score and context-FID, often by a large margin. We only observe a few isolated cases where the Predictive Score is slightly lower. Due to space limitations, we are unable to include the full per-dataset table here, but we will include it in the final revision.
>
> **Limitations 1**:  *The authors acknowledge some dataset-specific limitations in performance (e.g., on Weather and ECG200), but a deeper analysis into why the model underperforms on these datasets is missing ...*
>
> We made extensive efforts to understand why our model underperforms on datasets like Weather and ECG200, analyzing factors such as similarity to the pretraining distribution using mutual information and FFT-based metrics using correlation analysis with linear and non-linear tools. For example, Weather had an FFT profile close to the pretraining average, while ECG200 was much farther—yet both showed poor generation quality. Overall, these experiments did not yield a clear or consistent explanation. We will include these results in the final revision and plan to explore this further in future work.
>
> **Limitations 2**:  *while the model spans high-stakes domains like finance and biomedical data, the discussion on potential societal risks—such as misuse of synthetic data, overreliance on generated samples, or downstream model degradation due to hallucinations—is minimal*
>
> This is an important point. We will include a dedicated paragraph on this topic, and address it in the final version as follows:
>
> Our framework advances time-series generation by pre-training a unified model across multiple datasets, promising significant gains in data synthesis, especially in data-scarce scenarios. For example, using our approach, geophysicists can generate more high-fidelity examples of earthquake waveforms in locations with rare seismic events, improving hazard assessment models. In healthcare, researchers could generate synthetic, yet realistic, electronic health records to study disease progression without compromising patient privacy, or augment datasets for training diagnostic models for rare diseases where real-world data is limited.
>
> At the same time, because the model spans high-stakes domains like finance and biomedical data, its power to generate high-fidelity sequences also heightens significant societal risks. The potential for misuse of synthetic data is substantial; for instance, the same technology that helps medical researchers could be used to create fraudulent clinical trial data, potentially leading to the approval of ineffective treatments. Beyond direct misuse, a critical danger lies in the overreliance on generated samples, as models trained on augmented datasets may become less robust to the noisy, unpredictable nature of real-world data. Furthermore, there is a risk of downstream model degradation due to hallucinations—the generative model may produce data with subtle, unrealistic artifacts that, while plausible-looking, cause subsequent models to fail in unforeseen ways.

---

### Note · Authors · 2025-08-14

We thank all reviewers for their constructive feedback and recognition of our framework’s strengths: few-shot performance and cross-domain generalization (g6T2, zZSF), motivation (WSzz, 2Ztk), and contributions including DyConv (g6T2, WSzz), dataset tokens (g6T2, 2Ztk), dataset-label conditioning (WSzz), ImagenTime improvements, multi-domain pretraining, and our data-scarce benchmark (zZSF).

**g6T2**

Image/delay-embedding vs. sequence-native encodings - We added comparison with DiffusionTS and KoVAE when trained in unified way, showing our approach consistently outperforms sequence-native baselines in scarce-data settings.

We clarified cost: ≈8h on a single rtx4090 GPU; fine-tuning typically ~15 minutes on one GPU.

Clarified the transform is invertible/lossless, so generated sequences are fully interpretable

**WSzz**

We clarified the loss-weighting derivation (EDM/ImagenTime), dynamic masking procedure, and AdaGN with a concrete UNet block snippet.

We justified ImagenTime (NeurIPS’24), DiffTS (ICLR’24), KoVAE (ICLR’24), and added TimeVAE per feedback; we report that our model outperforms TimeVAE across setups.

We revised terminology to “time series generation of variable-length inputs” and “cross-temporal resolution transfer learning.”

Remaining concerns:
Despite addressing a critical, previously unstudied problem, novelty reservations remain.
TSGBench metrics (MDD, ACD, SD, KD, ED, DTW) were proposed only during discussion, not in the initial review, so adding them within the window wasn’t feasible.
Reviewer also argues some baselines are outdated; we respectfully disagree.

**zZSF**

Explained choice of short sequence (needed parity across diverse GAN/VAE/diffusion baselines); flagged long-sequence generation as future work.

We clarified train/test splits and averaging; committed to add t-SNE/PCA to sanity-check originality and to discuss sample-level audit metrics (Alaa et al.).

Reviewer confirmed all concerns were addressed.

**2Ztk**

We provided new experiment removes all financial data from pretraining; we still outperform ImagenTime on Stocks and Exchange across metrics.

We added TimeVAE (per request) and reported results.

Reviewer noted concerns were mostly addressed and raised the score.


Dataset token clarity & effectiveness (**WSzz, 2Ztk, g6T2**) - We clarified design and role; provided pretraining evidence, and a new ETT-family study shows clear gains when sampling with the token even under ambiguous boundaries.

---

### Decision · Program_Chairs · 2025-09-17

**Decision:**

Accept (poster)

**Comment:**

This paper introduces a unified generative modeling framework for time series generation, leveraging a vision-based diffusion model with DyConv layers and dataset-specific tokens to support diverse domains, variable sequence lengths, and robust few-shot performance.

The framework addresses the critical challenge of data scarcity, demonstrating strong results across finance, climate, energy, and biomedical datasets.

While some dataset-specific limitations remain, the authors have clarified methodology and evaluation details during the rebuttal. Overall, this is a well-motivated and timely contribution that can open new directions for time-series generative modeling research. I recommend acceptance.